# MAS-GPT: Training LLMs to Build LLM-based Multi-Agent Systems

Rui Ye [1 2 * @]   Shuo Tang [1 *]   Rui Ge [1]   Yaxin Du [1]   Zhenfei Yin [3 4]   Siheng Chen [1]   Jing Shao [2]

## Abstract

LLM-based multi-agent systems (MAS) have shown significant potential in tackling diverse tasks. However, to design effective MAS, existing approaches heavily rely on manual configurations or multiple calls of advanced LLMs, resulting in inadaptability and high inference costs. In this paper, we simplify the process of building an MAS by reframing it as a generative language task, where the input is a user query and the output is a corresponding MAS. To address this novel task, we unify the representation of MAS as executable code and propose a consistency-oriented data construction pipeline to create a high-quality dataset comprising coherent and consistent query-MAS pairs. Using this dataset, we train MAS-GPT, an open-source medium-sized LLM that is capable of generating query-adaptive MAS within a single LLM inference. The generated MAS can be seamlessly applied to process user queries and deliver high-quality responses. Extensive experiments on 9 benchmarks and 5 LLMs show that the proposed MAS-GPT consistently outperforms 10+ baseline MAS methods on diverse settings, indicating MAS-GPT's high effectiveness, efficiency and strong generalization ability. The codes are released at https://github.com/rui-ye/MAS-GPT.

## 1. Introduction

Large language models (LLMs) such as ChatGPT (Ouyang et al., 2022; OpenAI, 2023) have achieved significant success on a wide range of tasks. However, a single LLM often struggles to handle the diverse and complex range of tasks (e.g., varying difficulties and domains) encountered in practice (Hong et al., 2024; Chen et al., 2024).

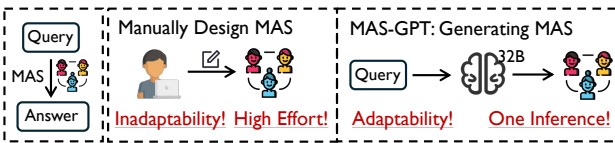

Figure 1: Introduction of our proposed new paradigm for building MAS. During inference, MAS-GPT adaptively generates a query-specific MAS with one LLM inference.

Such limitation has driven recent research towards building LLM-based multi-agent systems (MAS) (Ye et al., 2025a; Qian et al., 2024a; Chen et al., 2024), where multiple LLMs (agents) with specialized capabilities work collaboratively to achieve more effective solutions. For example, MetaGPT (Hong et al., 2024) and ChatDev (Qian et al., 2024a) build multi-LLM teams with expertise roles (e.g., programmer, tester, and product manager) to solve complex coding tasks in a predefined pipeline; while Agent-Verse (Chen et al., 2024) involves recruiters, executors, and evaluators for iterative task solving. These methods have shown superior performance over single LLM inference.

Despite achieving promising task performance, there are two fundamental issues that hinder the broad applications of MAS: inadaptability and high costs. (i) Inadaptability & high human effort: MAS in MetaGPT (Hong et al., 2024), ChatDev (Qian et al., 2024a), and AgentVerse (Chen et al., 2024) are all manually crafted (e.g., for coding tasks). That is, the collaboration structure and agents' prompts are predetermined and static, lacking in the generality to adapt towards any given tasks. (ii) High inference costs: Although there have been efforts to design adaptive MAS, they essentially shift the human cost onto the computational cost. For example, both GPTSwarm (Zhuge et al., 2024), AFlow (Zhang et al., 2024), and DyLAN (Liu et al., 2024b) rely on LLMs to replace human involvement, iteratively adjusting the collaboration structure or agents' prompts in the MAS for each specific task. However, this process often requires multiple LLM inferences and a corresponding validation set in advance (Zhuge et al., 2024).

Focusing on these key issues, this paper explores how to adaptively build a query-specific MAS at a minimal cost. Our core idea is to reframe the process of building an executable MAS for each query as a generative language task, making building MAS as simple and efficient as querying

---
[*]Primary contributor [@]Work done during Internship [1]Shanghai Jiao Tong University [2]Shanghai AI Laboratory [3]University of Oxford [4]The University of Sydney. Correspondence to: Siheng Chen <sihengc@sjtu.edu.cn>, Jing Shao <shaojing@pjlab.org.cn>.

*Proceedings of the 42nd International Conference on Machine Learning*, Vancouver, Canada. PMLR 267, 2025. Copyright 2025 by the author(s).

ChatGPT (Ouyang et al., 2022). Given the generated MAS, the query can then be seamlessly processed to produce the final response, significantly simplifying the whole pipeline.

Under this context, we introduce **MAS-GPT**, an LLM specifically trained to adaptively generate executable MAS based on any given user query in one single inference. While the concept is straightforward, the challenge lies in the limited knowledge of LLMs on the task of MAS generation and the lack of corresponding training data. These limitations raise two key technical challenges: *how to represent the MAS* and *how to construct the dataset.* (1) To ensure the generated MAS is readily executable, we unify the representation of MAS by describing it as a Python code snippet (i.e., a forward function), with each agent's prompt as a variable, LLM calls as functions, and agent relationships as string concatenation. (2) Building on this foundation, we propose a consistency-oriented data construction pipeline to facilitate the model in learning generalizable patterns and logical correlations, which includes the construction, evaluation, selection, and refinement of query-MAS pairs. During selection, we design an inter-consistency-oriented selection approach to ensure that similar queries are paired with similar high-performing MAS, facilitating the model to learn generalizable patterns. During refinement, we propose a intra-consistency-oriented refinement method to strengthen the relatedness between query and MAS, enabling the model to learn the reasonable correlation. Finally, the resulting pairs are used to train open-source LLMs via supervised fine-tuning, where the instruction is the user query and the response is the MAS represented by code. This will equip the model with the ability to generate query-specific MAS, and also, generalize to unseen queries.

With the introduction of MAS-GPT, inference for a query becomes significantly simplified. Instead of relying on manual crafting (Hong et al., 2024; Qian et al., 2024a; Chen et al., 2024) or multiple LLM inference costs (Liu et al., 2024b; Zhuge et al., 2024) to obtain an MAS for each query, the user simply inputs a query into MAS-GPT to get a corresponding executable MAS. Such MAS can be directly applied to process the query, where multiple MAS-GPT-generated agents collaborate with an MAS-GPT-generated structure to deliver the final solution. With advantages of adaptability, low cost, and generalization, this approach could facilitate the application of MAS at scale.

We conduct extensive experiments to compare MAS-GPT with 10+ baseline methods on 9 benchmarks (various domains) using 5 state-of-the-art (open-source and proprietary) LLMs. Our results show that MAS-GPT consistently outperforms baseline methods on average, indicating its high generality and effectiveness. Meanwhile, MAS-GPT has the potential to further push the boundary of strong reasoning capability of o1 (OpenAI, 2024b) and DeepSeek-R1 (Guo

et al., 2025), bringing 13.3% and 10.0% gain on AIME-2024 (a challenging mathematical benchmark), respectively. We also verify that our MAS-GPT can generalize to unfamiliar queries and generate novel MAS via case studies.

Our contributions are as follows:

1. We reframe building MAS for each query as a generative language task. We unify the representation of MAS as executable code and propose a consistency-oriented query-MAS data construction pipeline for LLM training.

2. We train MAS-GPT, an LLM that generates query-specific executable MAS within one single inference and can generalize across domains.

3. Experiments on 9 benchmarks and 5 LLMs show that MAS-GPT consistently outperforms 10+ baselines at a moderate cost during inference, indicating its effectiveness, efficiency and generalization ability.

## 2. Related Work

**LLM-based Multi-Agent Systems.** Since a single LLM may struggle to handle the diverse and complex range of tasks in practice (Li et al., 2023; Qian et al., 2024b), such limitation has driven recent research towards building LLM-based multi-agent systems (MAS) (Wang et al., 2024c; Wu et al., 2023). MetaGPT (Hong et al., 2024) and Chat-Dev (Qian et al., 2024a) introduce manually designed multi-agent teams for solving coding tasks; while MedAgents is designed for medical tasks (Tang et al., 2024). Agent-Verse (Chen et al., 2024) proposes an iterative collaboration structure where agents are recruited to discuss, execute, and evaluate. Multi-Agent Debate (Du et al., 2024; Liang et al., 2024) designs multiple expertise LLM-agents to debate and reason over multiple rounds to get final answers. The MAS in these methods are all fixed regardless of the given query, lacking in the generality to adapt accordingly.

DyLAN (Liu et al., 2024b) leverages LLMs to evaluate agents' values and dynamically select the best agents, GPTSwarm (Zhuge et al., 2024) manually initializes an agent team, adjusts the collaboration structure and agents' prompts by prompting LLMs. Given queries with ground-truth answers from one task and several available MAS as context, ADAS (Hu et al., 2024) and AFlow (Zhang et al., 2024) leverages the strong capabilities of LLMs such as Claude-3.5-sonnet (Anthropic, 2024) and GPT-4 (OpenAI, 2023) to iteratively generate task-oriented MAS for the specific task. Most of these methods require a corresponding validation set in advance (Zhuge et al., 2024; Zhang et al., 2024; 2025) and multiple times of LLM calls (e.g., over 10 calls of API with lengthy context) (Hu et al., 2024; Liu et al., 2024b) to obtain an MAS, which is compute-expensive and even unachievable in practice.

Instead of manually designing a fixed MAS (Qian et al., 2024a; Chen et al., 2024; Du et al., 2024; Ye et al., 2025b) or requiring multiple LLM inference costs to obtain an MAS (Liu et al., 2024b; Zhuge et al., 2024) for each query, our MAS-GPT significantly simplifies the process of building an MAS, which can flexibly generate query-specific MAS within one LLM inference. Meanwhile, unlike existing optimization-based methods (Hu et al., 2024; Zhang et al., 2024; Zhuge et al., 2024) that optimize on the validation set drawn from the domain same as the testing domain, our optimization (i.e., training) can be performed on diverse domains, enabling our MAS-GPT to generalize across domains. Specifically, we design a data-construction pipeline to generate a series of query-MAS pairs, which are used for training MAS-GPT based on open-source LLMs.

**LLM Post-Training.** Modern state-of-the-art LLMs are usually post-trained via two main stages: supervised fine-tuning (SFT) and preference learning (Ouyang et al., 2022; Dubey et al., 2024; Yang et al., 2024; Liu et al., 2024a), where SFT is the basic technique to teach LLM a defined tasks (Zhou et al., 2023; Longpre et al., 2023). Focusing on SFT, a series of researches are conducted on the construction of datasets for training chatbot-type LLMs. For example, LIMA (Zhou et al., 2023) manually annotates high-quality language data for SFT, emphasizing the importance of quality of SFT datasets. WizardLM (Xu et al., 2024), TULU 3 (Lambert et al., 2024), and Persona Hub (Ge et al., 2024) synthesize SFT data by prompting GPT models, indicating the potential of synthetic data for LLM training.

For MAS-GPT, the training process leverages SFT, with a primary focus on data construction. While previous approaches focus on training LLMs to directly answer user queries, the challenge of training LLMs to generate MAS from user queries introduces a novel difficulty. Unlike real-world dialogue data, LLMs have limited (if any) knowledge of MAS generation. Using our proposed data construction pipeline, we create the first query-MAS-paired dataset, facilitating the training of LLMs for MAS generation.

## 3. Methodology

This section first outlines the overall system integrated with MAS-GPT when processing user queries during inference. Next, we delve into the specifics of training MAS-GPT, with a particular focus on the dataset construction process.

### 3.1. Overall System Integrated with MAS-GPT

We follow a standard workflow: given a user query, a multi-agent system (MAS) is constructed, with multiple agents working collaboratively to generate the final answer. Unlike previous approaches that either manually design the MAS, rely on fixed and query-agnostic MAS, or incur significant

```python
def forward(query):
    math_agent = f'You are a math expert. Solve this\
        question: {query}'
    math_output = call_llm(math_agent)

    feedback_agent = f'Given {query} and {math_output},\
        provide feedback'
    feedback_output = call_llm(feedback_agent)

    refine_agent = f'Given {query}, {math_output} and \
        {feedback_output}, provide the final answer'

    return call_llm(refine_agent)
```

Figure 2: Our unified code representation of an executable MAS (i.e., a forward function). Each color denotes an agent. Agents defined by variables, LLM calls denoted by function calls, and interactions represented by string concatenations.

computational costs to determine the appropriate MAS, our approach streamlines the entire process of building MAS by reducing it to a single LLM inference.

The core of our system is MAS-GPT, an LLM that is trained to generate MAS tailored specifically to the input query. Instead of relying on pre-built agent configurations, MAS-GPT dynamically creates an MAS for each query, ensuring that the system adapts to a wide range of tasks. This approach not only minimizes the time and computational resources traditionally required to build the right MAS but also enhances the system's flexibility by generating task-specific solutions in real-time. Finally, the MAS generated by MAS-GPT can be seamlessly integrated to process the query and deliver the final answer (bottom right in Figure 3).

### 3.2. MAS-GPT: Dataset Construction and Training

To achieve the above goal, we reframe building MAS as a generative language task, where the input is a user query and the output is an executable MAS capable of processing that query. This shift to a generative paradigm introduces a new challenge since there is few (if any) knowledge within LLMs on MAS generation. To make this approach viable, the key focus lies in constructing an appropriate dataset to teach the LLMs such brand-new task. To achieve this, we propose a consistency-oriented data construction process, which involves four key steps: (1) construction of query and MAS pools, (2) inference and evaluation of query-MAS pairs, (3) inter-consistency-oriented pair selection, (4) intra-consistency-oriented pair refinement.

**Data - Construction of Query and MAS Pools (Representing MAS as Executable Code).** To construct the dataset for supervised fine-tuning (SFT), we adopt the following data format: *(system prompt, instruction, response)*. Here, the system prompt briefly describes the MAS generation task, the instruction corresponds to the user query, and the response includes the MAS, which can be extracted by string matching. Therefore, training the LLM requires the collection of a series of query-MAS pairs. Firstly, to enable

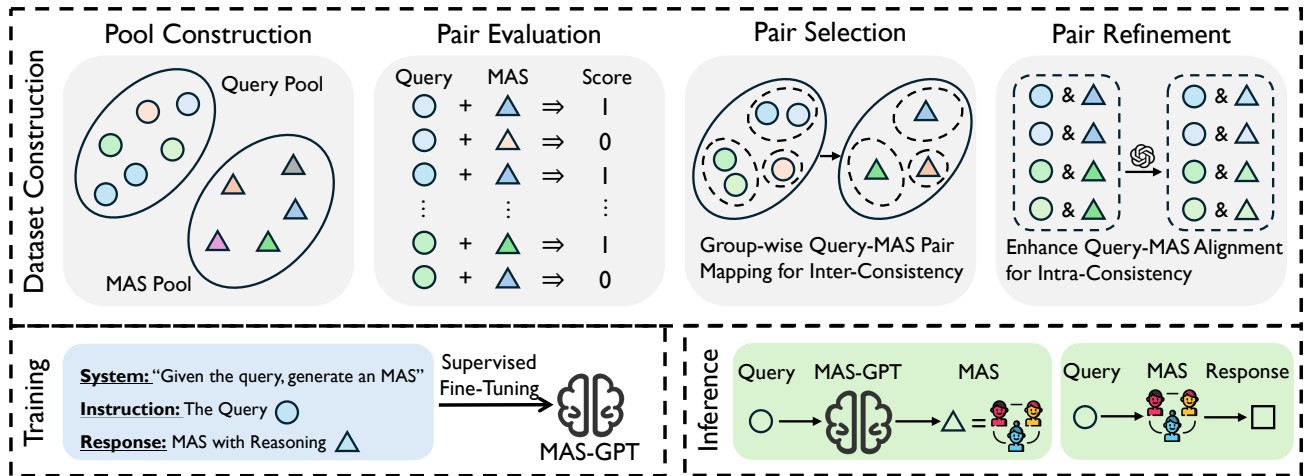

Figure 3: Illustrations of dataset construction, training, and inference of our MAS-GPT. After training (which is one-time), the inference is significantly simplified: MAS-GPT generates a query-specific MAS, which can directly process the query.

MAS-GPT to handle diverse queries, we build a query pool from open-source queries across various domains, such as general QA, mathematics, and coding. Each query is carefully selected to be verifiable, ensuring the presence of a ground-truth answer or test cases (e.g., for coding tasks).

While the collection of queries is relatively straightforward, constructing the MAS pool presents a fundamental challenge: *how to represent an executable MAS*. To address this, we propose unifying the representation of MAS by formalizing it as executable Python code snippets. This unified representation is motivated by the observation that all existing LLM-based MAS methods are ultimately implemented as code, encompassing the definition of agents' prompts, LLM calls, and inter-agent interactions (Qian et al., 2024a; Hu et al., 2024; Zhang et al., 2024). Specifically, we define an MAS as a forward function that takes a user query as input and returns the final answer generated by the MAS. Within the forward function, agent prompts are defined as variables, agent inferences are implemented as function calls, and interactions between agents are represented through string concatenation; see an example in Figure 2.

Following this framework, we first re-implement several existing MAS methods (e.g., Multi-Agent Debate (Du et al., 2024), Self-Consistency (Wang et al., 2024b), Self-Refine (Madaan et al., 2024)) to align with our unified code representation. To further expand the diversity of MAS candidates, we also manually design some MAS systems, resulting in a base MAS pool comprising over 40 unique MAS designs (Figure 6). These 40 MAS cover basic elements such as chain-of-thought prompts, role-playing prompts, LLM-calling functions, and functions to get code execution results. As the base LLM does not know our representation of MAS (verified by poor performance in Figure 5(b)), including these basic elements is critical to teach the LLM

our MAS representation. Importantly, these 40+ MAS do not directly correspond to the exact number of MAS in the training dataset; rather, they serve as foundations that evolve during the query-MAS pair refinement process. See more details about implementation and visualization in Section A.

**Data - Evaluation of Query-MAS Pairs.** After constructing the query and MAS pools, it is crucial to evaluate the query-MAS compatibility since not all MAS designs are equally suitable for every query. To achieve this, we pair each query and MAS in the pool by inferring the query to the MAS and evaluating the generated final answer.

Specifically, given the query pool $\mathbf{Q} = \{(Q_i, Y_i)\}_{i=1}^N$ and the base MAS pool $\mathbf{M} = \{MAS_j^{\text{base}}\}_{j=1}^M$, where $Q_i$ is the query, $Y_i$ is the information for verification, $N$ and $M$ denotes the pool size, we obtain $N \times M$ pairs. Then, a query-dependent evaluation function $f_{\text{eval}}(\cdot)$ will be applied to evaluate the effectiveness of the query-MAS pair: $score_{i,j} = f_{\text{eval}}(MAS_j^{\text{base}}(Q_i), Y_i)$, where $MAS_j^{\text{base}}(Q_i)$ denotes the answer generated by $MAS_j^{\text{base}}$ given the query $Q_i$, 1 and 0 denotes correct and wrong respectively. Overall, we get $M$ MAS scores for each query $Q_i$, which are denoted by $\mathbf{s}_i = [score_{i,1}, ..., score_{i,M}]$, laying the foundation for subsequent steps for selecting appropriate query-MAS pairs and further refinement.

**Data - Inter-Consistency-Oriented Pair Selection.** With the query-MAS pair results obtained from the evaluation step, the next critical task is to select and construct high-quality query-MAS pairs for training. The first selection criterion is intuitive: *effectiveness*. Specifically, we retain only the query-MAS pairs where the MAS produces a correct answer (evaluation score is 1), as MAS designs that generate correct answers are more likely to be suitable for their respective queries compared to those that fail.

While using all the remaining effective query-MAS pairs for training is straightforward, this approach introduces a significant problem of low *inter-consistency*: the same or similar queries may correspond to multiple different MAS designs. This lack of consistency makes it difficult for the model to learn a clear optimization objective, hindering its ability to understand and perform the task effectively.

To address this issue, we propose an inter-consistency-oriented pair selection method that optimizes both *effectiveness* and *inter-consistency*. The core idea is to group similar queries and assign them a single, high-performing MAS to maintain consistency across the dataset. Specifically, we cluster queries based on their metadata or embeddings. For a group of $S$ queries $\mathbb{S} = \{Q_i\}_{i=1}^{S}$, we calculate a cumulative score for each MAS by summing its effectiveness scores across all queries in the group: $\mathbf{s} = \sum_{i=1}^{S} \mathbf{s}_i$. The MAS with the highest cumulative score is then selected as the representative MAS for all queries in the group: $MAS_*^{\text{base}} = \arg\max_{MAS \in \mathbf{M}} \mathbf{s}$. Through this, we pair each specific query with a specific base MAS: $(Q_i, MAS_i^{\text{base}})$.

By aligning similar queries with the same high-performing MAS, this method improves the inter-consistency of the query-MAS pairs, facilitating the model in learning to recognize generalizable patterns and generalize across similar queries. For example, queries requiring divergent thinking may be consistently paired with MAS structures where multiple agents independently generate ideas and then discuss.

**Data - Intra-Consistency-Oriented Pair Refinement.** While the inter-consistency-oriented pair selection process effectively ensures consistency across query-MAS pairs, there remains a critical issue within individual pairs: *intra-consistency*. Specifically, the alignment between a query and its associated MAS may still be suboptimal, making it challenging for the model to learn meaningful associations. For instance, a query about physics may be paired with an MAS involving experts from multiple domains (e.g., physics and biology), where the presence of non-relevant agents like biology experts can confuse the model.

To address this, we propose an *intra-consistency*-oriented pair refinement method. This approach aims to improve the query-MAS alignment through two key strategies: (1) adjusting MAS to make it query-dependent, and (2) introducing a reasoning process to strengthen the connection between the query and MAS. We employ an LLM-based data synthesis method, where an advanced LLM adjusts agents' definitions within the MAS based on the query and the previously selected MAS. The LLM is also instructed to generate a reasoning statement that explains the relationship between the query and the refined MAS, improving the interpretability of the query-MAS pair; please refer to prompt in Table 11. This process enables the model to better understand the context and logic behind each decision, which in

turn facilitates model training and improves generalization.

Next, we infer and evaluate the refined MAS on the corresponding query, as advanced LLMs could generate inappropriate or non-executable MAS. Specifically, for each base pair $(Q_i, MAS_i^{\text{base}})$, the refined $MAS_i^{\text{refine}}$ is tested and accepted only if it achieves a not-worse score. Formally:

$$MAS_i = \begin{cases} MAS_i^{\text{refine}}, & \text{if } s^{\text{refine}} \geq s^{\text{base}} \\ MAS_i^{\text{base}}, & \text{otherwise} \end{cases},$$

where $s^{\text{refine}} = f_{\text{eval}}(MAS_i^{\text{refine}}(Q_i), Y_i)$ and $s^{\text{base}} = f_{\text{eval}}(MAS_i^{\text{base}}(Q_i), Y_i)$ are evaluation scores by comparing the MAS-generated and ground-truth answer $Y_i$.

Through this process, each query $Q_i$ is ultimately associated with a tuple $(Q_i, R_i, MAS_i)$, where $R_i$ denotes the reasoning statement, and $MAS_i$ is the final MAS. This refined dataset ensures both inter- and intra-consistency, providing high-quality training data for subsequent model fine-tuning.

**Training - Supervised Fine-Tuning of MAS-GPT** Our dataset follows the format *(system prompt, instruction, response)*. The system prompt briefly describes the task of generating a query-specific MAS and the instruction corresponds to the user query $Q_i$. The response is constructed as the concatenation of the reasoning process and the final MAS, which is represented as executable code in text form.

Building upon this dataset, we perform supervised fine-tuning of MAS-GPT on the open-source medium-sized LLM, Qwen2.5-Coder-32B-Instruct (Yang et al., 2024), leveraging its capabilities of code generation and instruction-following. During inference, when a user query is received, MAS-GPT generates an executable MAS tailored to that specific query $Q_i$: $MAS_i^{\text{gen}} = \text{MAS-GPT}(Q_i)$. The generated MAS is directly usable for processing the query $Q_i$ and delivering the final answer: $A_i = MAS_i^{\text{gen}}(Q_i)$, significantly simplifying the task handling process.

### 3.3. Discussions

**Advantages.** Overall, our system integrated with MAS-GPT offers the following key advantages: simplicity, cost-efficiency, and adaptability (generality). Instead of manually designing an MAS for each specific query, relying on a fixed MAS for all queries, or requiring multiple LLM inference costs to obtain an MAS for a query, our MAS-GPT significantly simplifies the process of building an MAS by reducing into one single LLM inference. Given a user query, MAS-GPT will efficiently return a query-specific MAS, which is executable and can be seamlessly applied to process the query to deliver the final answer. Although training incurs some cost, it is a one-time expense, whereas inference is potentially endless in practical applications. We believe that MAS-GPT has the potential to further advance the real-world application of MAS due to its simplicity,

Table 1: Comparing MAS-GPT with 10 baselines across 8 benchmarks using Llama-3-70B-Instruct, MAS-GPT performs the best on average, verifying its generality in handling diverse queries. Benchmarks with * are out-of-domain for MAS-GPT.

| METHOD | MATH | GSM8K | GSM-H | H-EVAL* | H-EVAL+* | MMLU | GPQA* | SCIBENCH* | AVG. |
|---|---|---|---|---|---|---|---|---|---|
| SINGLE (DUBEY ET AL., 2024) | 50.55 | 92.38 | 45.80 | 79.01 | 75.78 | 77.37 | 36.68 | 21.05 | 59.83 |
| CHAIN-OF-THOUGHT (WEI ET AL., 2022) | 53.20 | 92.79 | 46.20 | 77.16 | 77.02 | 75.56 | 35.28 | 17.68 | 59.36 |
| SELF-CONSISTENCY (WANG ET AL., 2024B) | 61.59 | 94.99 | 47.20 | 77.78 | 75.78 | 78.18 | 37.15 | 20.00 | 61.58 |
| LLM-DEBATE (DU ET AL., 2024) | 61.37 | 91.58 | 44.60 | 74.69 | 74.53 | 77.78 | 34.35 | 19.79 | 59.84 |
| SELF-REFINE (MADAAN ET AL., 2024) | 58.50 | 90.78 | 37.80 | 67.90 | 62.73 | 74.75 | 38.32 | 20.00 | 56.35 |
| QUALITY-DIVERSITY (LU ET AL., 2024) | 60.49 | 92.99 | 45.60 | 70.99 | 70.19 | 75.76 | 33.64 | 20.63 | 58.79 |
| SPP (WANG ET AL., 2024C) | 51.66 | 92.79 | 44.80 | 76.54 | 73.29 | 77.37 | 35.05 | 20.84 | 59.04 |
| AGENTVERSE (CHEN ET AL., 2024) | 55.63 | 93.39 | 41.40 | 77.78 | 73.91 | 76.57 | 40.19 | 16.00 | 59.36 |
| GPTSWARM (ZHUGE ET AL., 2024) | 55.41 | 93.19 | 43.20 | 69.14 | 73.91 | 75.15 | 36.45 | 14.11 | 57.57 |
| DYLAN (LIU ET AL., 2024B) | 59.60 | 91.18 | 44.80 | 79.01 | 75.78 | 78.18 | 35.98 | 19.79 | 60.54 |
| **MAS-GPT (OURS)** | 68.65 | 93.39 | 62.40 | 80.25 | 78.88 | 78.38 | 37.62 | 24.21 | **65.47** |

Table 2: Statistics of MAS-GPT's training dataset. We show the number of samples $N_{data}$; the averaged instruction $L_{ins}$; the averaged response $L_{res}$, reasoning $L_{rsn}$, and MAS length $L_{MAS}$; and the number of unique MAS $N_{MAS}$.

| $N_{data}$ | $L_{ins}$ | $L_{res}$ | $L_{rsn}$ | $L_{MAS}$ | $N_{MAS}$ |
|---|---|---|---|---|---|
| 11442 | $\sim 75.0$ | $\sim 1062.3$ | $\sim 262.5$ | $\sim 784.8$ | 7580 |

cost-efficiency, and adaptability.

**Cost.** While MAS-GPT simplifies the application of MAS during inference time, it incurs cost for collecting data and training LLMs. However, training is one-time while inference could be endless (similarly OpenAI trains GPT-4 for one time and serves the world countless times). The MAS-GPT is trained on diverse data only once, and then can generalize to diverse domains for wide applications. Training-time efficiency is not the focus of this paper, which could be a potential future direction.

We believe that this paradigm is scalable due to an exciting and promising property of MAS-GPT: it could stand on the shoulders of giants. Ideally, if we could include all existing high-performance MAS methods into the MAS pool, in real-world applications, we only need to deploy MAS-GPT to solve diverse user queries, rather than deploying multiple MAS and designing complicated rules to select them.

## 4. Experiments

### 4.1. Experimental Setups

**Training.** Our training queries are sampled from the training splits available in MATH (Hendrycks et al., 2021b), GSM8K (Cobbe et al., 2021), MBPP (Austin et al., 2021), MMLU (Hendrycks et al., 2021a), and SciQ (Welbl et al., 2017), covering domains of math, coding, and general QA. Llama-3-70B-Instruct is used during dataset construction. The number of training samples (i.e., query-MAS pairs) is approximately 11k. We report the statistics of our dataset in Table 2, where the number of unique MAS is measured by string comparisons. Our MAS-GPT is trained over

Qwen2.5-Coder-32B-Instruct (Yang et al., 2024), leveraging its instruction-following and coding capabilities. We train the LLM using 16 A100s with an effective batch size of 32 for 3 epochs at a learning rate of 1e-5 (Zheng et al., 2024).

**Testing.** To verify that our MAS-GPT can handle diverse queries in practice, we consider multiple benchmarks from diverse domains. These include MATH (Hendrycks et al., 2021b), GSM8K (Cobbe et al., 2021), GSM-Hard (Gao et al., 2023), AIME-2024 for math domains; HumanEval (Chen et al., 2021) and HumanEval+ (Liu et al., 2023) for coding tasks; MMLU (Hendrycks et al., 2021a) for general QA tasks; GPQA (Rein et al., 2023) and SciBench (Wang et al., 2024a) for science topics. Among these, AIME-2024, HumanEval, HumanEval+, GPQA, and SciBench are from a totally different distributions compared to training data, serving to verify the generalization capability of MAS-GPT. While there are samples related to mathematics and science during training, during testing, the problems in AIME, GPQA, and SciBench are much more challenging. Please refer to Table 10 for details about datasets and Section C.2 for details about evaluation. For all baselines, the LLMs that drive the MAS to process user queries are kept the same, where we consider five state-of-the-art LLMs including Llama-3-70B-Instruct (Dubey et al., 2024), Qwen2.5-72B-Instruct (Yang et al., 2024), GPT-4o-mini-2024-07-18 (OpenAI, 2024a), o1-preview-2024-09-12 (OpenAI, 2024b), and Deepseek-R1 (Guo et al., 2025).

**Baselines.** For fair comparisons, we consider 11 baselines that are suitable for handling diverse tasks. We include single agent and agent with chain-of-thought (Wei et al., 2022) as two basic baselines, Self-Consistency (Wang et al., 2024b) and Quality-Diversity (Lu et al., 2024) that select the best from multiple answers, LLM-Debate (Du et al., 2024) that involves multiple experts for debating, Self-Refine (Madaan et al., 2024) that iteratively refines last agent's answer, SPP (Wang et al., 2024c) that stimulates conversations among multiple roles, AgentVerse (Chen et al., 2024) and DyLAN (Liu et al., 2024b) that dynamically adjust multi-agent team during inference, GPTSwarm (Zhuge et al., 2024) that relies on a graph collaboration structure.

We also compare with an MAS optimized by AFlow (Zhang et al., 2024) on math task.

### 4.2. Main Results

Since our proposed MAS-GPT aims to facilitate the multi-agent systems in flexibly handling diverse queries, our results focus on the keyword of generality. Here, we show the generality of MAS-GPT by comparing performance averaged on various benchmarks and performance using different LLMs to drive the MAS.

**MAS-GPT's generality in handling diverse queries.** We compare MAS-GPT with 10 baselines on 8 benchmarks using Llama-3-70B-Instruct (Dubey et al., 2024) to drive the MAS, with results reported in Table 1. GPQA and SciBench are two benchmarks that are out-of-domain for our MAS-GPT. From the table, we see that (1) our MAS-GPT significantly outperforms the baseline methods on average, outperforming the second-best method by 3.89%. (2) Our MAS-GPT simultaneously achieves promising performance in both in-domain and out-of-domain (i.e., queries that are significantly different from those in the training data) benchmarks, indicating MAS-GPT's generality. (3) MAS-GPT is the only method that consistently achieves better performance than single agent, indicating its robustness in handling diverse queries.

**Generality in using diverse LLM backbones for MAS.** Llama-3-70B-Instruct was utilized to drive MAS during the dataset construction phase for training MAS-GPT, a 32B-sized LLM. As shown in Table 1, this approach proves effective when employing the same LLM to drive MAS during test time. To further validate the versatility of MAS-GPT, we assess its performance under different MAS-driving LLMs, including Qwen2.5-72B-Instruct and GPT-4o-mini-2024-07-18, in Table 3. The results demonstrate that MAS-GPT consistently achieves superior performance, regardless of the LLM used to drive MAS, highlighting its strong compatibility and adaptability across various MAS-driving LLMs.

**MAS-GPT's potential in further augmenting the reasoning performance of strong reasoning LLMs such as o1.** In recent developments, the AI community has introduced several state-of-the-art reasoning LLMs (OpenAI, 2024b; Qwen, 2024), which have demonstrated remarkable reasoning capabilities by scaling inference-time computations (Snell et al., 2024). In this context, we aim to explore whether our proposed MAS-GPT can take the reasoning power of these already advanced models even further. To test this, we conduct experiments using OpenAI's o1-preview (OpenAI, 2024b) and Deepseek-R1 (Guo et al., 2025), evaluating them on the highly challenging AIME-2024 mathematical benchmark[1]. The results, as shown in

[1] https://huggingface.co/datasets/Maxwell-Jia/AIME_2024

Table 3: MAS-GPT consistently performs the best across MAS-driving LLMs, indicating its strong compatibility.

| METHOD | MATH | GSM-H | H-EVAL+ | MMLU | GPQA | AVG. |
|---|---|---|---|---|---|---|
| **QWEN2.5-72B-INSTRUCT** | | | | | | |
| SINGLE | 85.86 | 64.91 | 85.37 | 82.60 | 44.39 | 72.63 |
| COT | 86.90 | 62.27 | 84.15 | 83.20 | 47.86 | 72.88 |
| SELF-CON. | 87.32 | 61.46 | 87.20 | 83.40 | 50.00 | 73.88 |
| LLM-DEBATE | 85.24 | 63.49 | 68.90 | 86.20 | 47.86 | 70.34 |
| SELF-REFINE | 83.58 | 59.03 | 78.66 | 85.40 | 43.32 | 70.00 |
| Q-D | 85.65 | 63.08 | 76.83 | 82.80 | 48.66 | 71.40 |
| SPP | 85.65 | 62.88 | 82.32 | 83.40 | 48.40 | 72.53 |
| AGENTVERSE | 84.82 | 59.43 | 81.10 | 83.20 | 44.65 | 70.64 |
| GPTSWARM | 83.16 | 63.89 | 83.54 | 84.60 | 44.92 | 72.02 |
| DYLAN | 87.73 | 63.08 | 85.37 | 84.40 | 51.07 | 74.33 |
| **MAS-GPT** | 87.53 | 66.33 | 85.98 | 83.80 | 48.66 | **74.46** |
| **GPT-4O-MINI-2024-07-18** | | | | | | |
| SINGLE | 78.18 | 58.03 | 86.25 | 78.56 | 38.03 | 67.81 |
| COT | 78.79 | 60.84 | 85.62 | 79.16 | 39.60 | 68.80 |
| SELF-CON. | 81.62 | 59.04 | 85.00 | 80.96 | 39.82 | 69.29 |
| LLM-DEBATE | 79.60 | 60.84 | 86.25 | 80.76 | 37.81 | 69.05 |
| SELF-REFINE | 74.55 | 54.62 | 76.88 | 79.16 | 33.33 | 63.71 |
| Q-D | 79.80 | 59.64 | 84.38 | 79.76 | 37.58 | 68.23 |
| SPP | 77.58 | 57.63 | 86.25 | 77.96 | 37.58 | 67.40 |
| AGENTVERSE | 75.15 | 55.62 | 79.38 | 78.36 | 36.24 | 64.95 |
| GPTSWARM | 75.15 | 55.62 | 79.38 | 78.36 | 36.32 | 64.97 |
| DYLAN | 81.21 | 59.24 | 80.62 | 79.96 | 40.94 | 68.39 |
| **MAS-GPT** | 81.21 | 61.45 | 86.88 | 80.36 | 42.60 | **70.50** |

Figure 4(a), show that our proposed MAS-GPT significantly outperforms the baseline methods on this challenging task. Specifically, it can improve over the single LLM by a large margin: **13.34%**. This result not only verifies the generality of our proposed MAS-GPT, but also indicates its promising potential in pushing the boundaries of LLM reasoning.

**Comparisons with task-specific methods, AFlow.** To further demonstrate the generality and effectiveness of our MAS-GPT during inference time, we compare with AFlow (Zhang et al., 2024), a latest task-specific method for MAS optimization that has been specifically optimized on MATH (Hendrycks et al., 2021b) dataset. We evaluate on two AFlow's in-domain (MATH and GSM8K) and two AFlow's out-domain (MMLU and HumanEval+) benchmarks. Results in Figure 4(b) show surprisingly good performance of our proposed MAS-GPT. As a general method, our *MAS-GPT even outperforms math-specific AFlow on the MATH* dataset by 3.53%! Meanwhile, the MAS optimized on MATH by AFlow fails to generalize to other domains, achieving worse performance than a single LLM. In contrast, our MAS-GPT consistently performs the best across these benchmarks. It is also worth mentioning that our MAS-GPT only requires *one-time inference of a 32B-sized LLM* to build the MAS; while AFlow needs to call the APIs of powerful proprietary LLMs, such as Claude-3.5-Sonnet (Anthropic, 2024), 10 times per query and depends on a hold-out validation set.

**Cost comparisons.** Here, we compare the inference cost of various methods from the moment a user query is received to the generation of the final answer, as illustrated in Figure 4(c). We quantify the inference cost in terms of the number of LLM inference calls (Liu et al., 2024b), interpret-

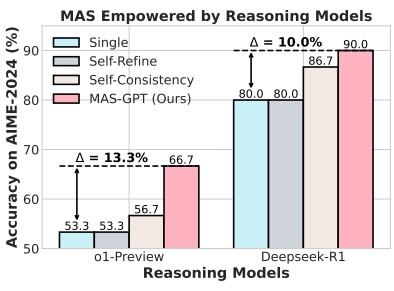

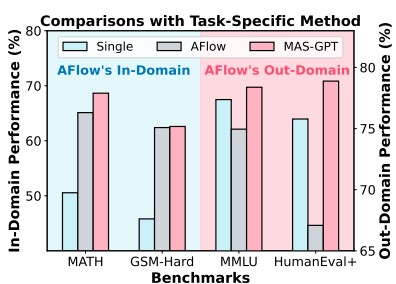

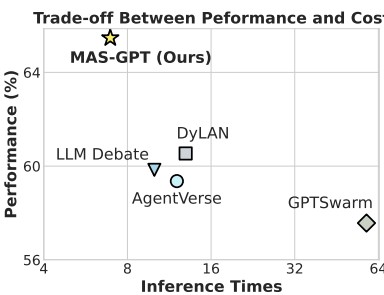

(a) MAS-GPT-Assisted Reasoning     (b) Comparison with Task-Specific Method     (c) Performance v.s. Inference Times

Figure 4: (a) Different methods empowered with strong reasoning LLM: o1-preview. We see that our MAS-GPT significantly enhance the reasoning performance over single LLM, indicating its potential in further augmenting LLM reasoning. (b) Comparisons with AFlow (optimized on MATH). MAS-GPT even outperforms AFlow on its in-domain benchmarks; while AFlow fails on out-of-domain benchmarks. (c) MAS-GPT achieves the best performance with low inference cost.

Table 4: Ablation studies on the designs of dataset construction: (1) our inter-consistency-oriented pair selection, (2) the adjustment of MAS in our intra-consistency-oriented pair refinement: Refine-A, (3) the introduction of reasoning process in our intra-consistency-oriented pair refinement: Refine-R. The table shows that these three designs all play critical roles in achieving high task performance.

| | SELECT | REFINE-A | REFINE-R | MATH | MMLU | GPQA |
|---|---|---|---|---|---|---|
| ① | ✗ | ✓ | ✓ | 60.26 | 77.58 | 36.68 |
| ② | ✓ | ✗ | ✓ | 66.23 | 77.78 | 36.45 |
| ③ | ✓ | ✓ | ✗ | 64.90 | 75.96 | 37.15 |
| ④ | ✓ | ✓ | ✓ | **68.65** | **78.38** | **37.62** |

ing the inference of MAS-GPT as 0.5 times, given that its model size is approximately half that of the MAS-driving LLM (32B v.s. 70B). From the figure, we observe that, among the four methods compared, MAS-GPT achieves the best performance while requiring the fewest inference calls, demonstrating its efficiency and effectiveness.

### 4.3. Analysis of MAS-GPT

**Effectiveness of inter-consistency-oriented pair selection.** During data construction, to facilitate the model in recognizing generalizable patterns between queries and MAS, we propose an inter-consistency-oriented query-MAS pair selection method, which maps similar queries with consistent high-performing MAS. To examine its effectiveness, we replace this mapping with a random mapping approach, which randomly selects one out of those MAS with correct answers. From Table 4, by comparing ① and ④, we see that our proposed method brings significant performance gain, with an absolute improvement of 8.39% on MATH.

**Effectiveness of intra-consistency-oriented pair refinement.** During data construction, to help the model learn the associations between query and MAS, we propose an intra-

consistency-oriented query-MAS pair refinement method. This method enhances the alignment between query and MAS by adjusting MAS to make it query-dependent and introducing a reasoning process to strengthen the logical connection. To examine their effects, we conduct two experiments with one without adjustment of MAS and one without reasoning process. From Table 4, by comparing ② and ④, ③ and ④, we see that our designs in adjusting MAS and introducing reasoning process both contribute to performance improvement, indicating the effectiveness of our proposed intra-consistency-oriented pair refinement.

**Scaling effects of data size.** To explore the scaling effects of data size for training MAS-GPT, we adjust the size from 0 to 11k using the same 32B-sized model and compare the extractability (i.e., the Python code can be extracted), executability (i.e., the code is executable), task performance. Results in Figure 5(a) show that except for the extractability under 0 data sample (the base model knows that it needs to generate Python code, but do not know what codes are needed), the extractability and executability generally improves with the data scale. Results in Figure 5(b) show (1) the base model is unable to generate an effective MAS in zero-shot setting, indicating the necessity for training MAS-GPT. Overall, we observe a promising scaling trend of training MAS-GPT: more data leads to better performance.

**Scaling effects of model size.** Here, we compare the performance of MAS-GPT trained based on 7B, 14B, and 32B models. Results in Figure 5(c) show that the performance of MAS-GPT improves steadily with the growing model size. Overall, these findings demonstrate the promising potential of MAS-GPT, suggesting that it can be further improved with more diverse, high-quality data and stronger models as the community continues to advance.

**Examining the generated MAS.** To ensure that our MAS-GPT is not simply memorizing those MAS seen during the

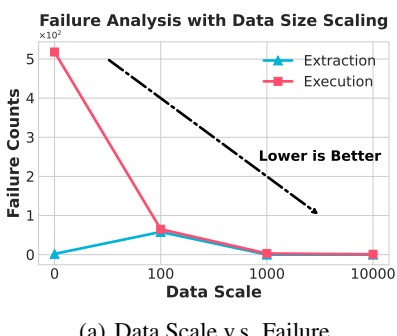

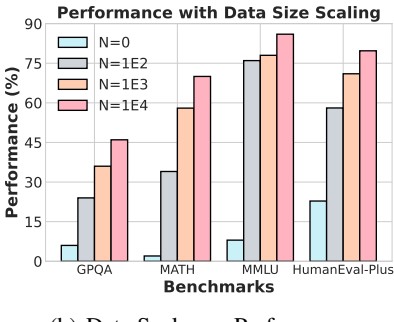

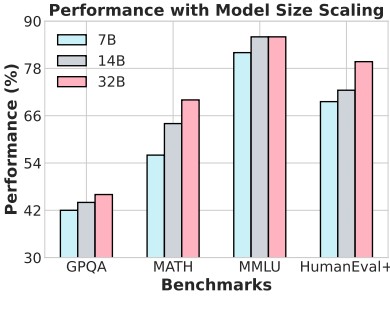

(a) Data Scale v.s. Failure      (b) Data Scale v.s. Performance      (c) Model Scale v.s. Performance

Figure 5: Explorations of scaling in training MAS-GPT. (a) More data leads to fewer execution failures. (b) More data contributes to better performance of MAS-GPT in facilitating MAS application. Without training (N=0), the model fails, highlighting that MAS generation is a non-trivial task requiring specific training. (c) Larger model generally contributes to better performance. These findings demonstrate the promising potential of MAS-GPT, suggesting that it can be further improved with more diverse, high-quality data and stronger models as the community continues to advance.

Table 5: The proportions of generated MAS that are novel compared to those in the training set. We see that MAS-GPT is not simply memorizing MAS seen during training; rather it learns to generate appropriate query-dependent MAS.

| BENCHMARK | GSM-H | MATH | SCIBENCH | MMLU |
|---|---|---|---|---|
| NOVEL (%) | 92.5 | 69.0 | 71.0 | 42.9 |

training process, we compare the generated MAS given queries during testing with those in the training set. We report the proportion of novel MAS in Table 5. From the table, we see that our MAS-GPT indeed generates novel MAS during test time. The proportions are not consistent across benchmarks, which may stem from the varying diversity of MAS associated with different domains during training.

To offer an intuitive understanding, we present several examples in Appendix showcasing the query, the MAS-GPT-generated reasoning process, and the MAS-GPT-generated MAS. These show that MAS-GPT can generate query-specific MAS (Section B.1), generalize to unseen queries (Section B.2), generate novel MAS (Section B.3).

## 5. Conclusions

Building MAS was time-consuming and resource-intensive. This paper aims to streamline this process into a single LLM inference, making MAS creation as effortless as querying ChatGPT. To this end, we introduce MAS-GPT, an LLM specifically trained to generate executable MAS from arbitrary user queries. Our approach follows a data-driven spirit, leveraging a consistency-oriented data construction pipeline to enhance the coherence and consistency of data pairs. We conduct extensive experiments, comparing MAS-GPT against 10+ baseline methods across 9 benchmarks, using 5 different LLMs as MAS drivers. The results consis-

tently demonstrate that MAS-GPT outperforms all baselines, strongly validating its effectiveness and generalizability. Additionally, we observe MAS-GPT's potential to further enhance state-of-the-art reasoning capabilities, as well as its scalability for continued improvements. We believe MAS-GPT can accelerate the adoption of MAS, inspiring future research and real-world applications.

*Limitations and Future Works.* While MAS-GPT shows promising results, several limitations and avenues for future work remain. First, constrained by limited human annotation and computational resources, the diversity of our initial MAS library and the final training set, potentially biased by mapping multiple queries to strong-performing MAS instances, could be further improved. Second, beyond code execution, integrating additional tools like multi-modal data processing and web search tools could significantly enhance MAS-GPT's capabilities. Third, as a foundational step, our current work utilizes only supervised fine-tuning (SFT); future research could explore reinforcement learning (RL) methods to enable MAS-GPT to autonomously explore and refine MAS generation. We believe that there is substantial room to explore along this direction, continuously advancing the flexibility of MAS in real-world applications.

## Impact Statement

This paper presents an advancement in simplifying the application of LLM-based multi-agent systems (MAS). Our MAS-GPT significantly reduces the complexity of designing and deploying MAS for a wide range of tasks. Our approach not only makes MAS development more accessible and efficient but also improves scalability, enabling its broader application in real-world scenarios. While the potential negative impacts of our approach are similar to those associated with large language models, such as eth-

ical concerns and misuse, these are inherent to the use of LLMs in general and do not require further elaboration in this context.

## Acknowledgments

This research is supported by the National Natural Science Foundation of China under Grant 62450162 and Grant 62171276; and Shanghai Artificial Intelligence Laboratory.

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

# A. Details of The Initial MAS Pool

## A.1. Implementations

We totally implement 40+ MAS in our initial MAS pool, which covers basic elements such as chain-of-thought prompts, role-playing prompts, LLM-calling functions, and functions to get code execution results. As the base LLM does not know our representation of MAS (verified by poor performance in Figure 5(b), N=0), including these basic elements is critical to teach the LLM our MAS representation.

**Basic functions:**

1. *call_llm(prompt)*, which takes the prompt as input and outputs the response generated by the LLM.

2. *execute_code(code_text)*, which takes a code snippet as input and outputs the execution results (including printed information or errors).

3. *test_code_get_feedback(code, test_cases)*, which tests the given code with the test cases and returns the feedback.

4. *get_function_signature(llm, taskInfo)*, which calls the LLM to return the generated function signature for the given task.

5. *get_test_cases(llm, taskInfo, function_signature)*, which calls the LLM to return the generated test cases for the given task and function signature.

6. *extract_code_solution(solution)*, which returns the code by extracting (wrapped within ¡Code Solution¿ and ¡/Code Solution¿) from the given solution.

7. *generate_and_extract_code(llm, prompt)*, which calls the LLM to return the generated response and the extracted code from the response (based on *extract_code_solution(solution)*).

Representing MAS as python code is a promising direction, as most functionalities in our AI community can be represented by python code. Along this new direction, we believe that with more data, tools, MAS, MAS-GPT could be continually improved.

## A.2. Visualization of Our MAS pool

We visualize several MAS in our MAS pool in Figure 6.

# B. Case Study

## B.1. MAS-GPT Can Generate Query-Specific MAS

From Table 6, we see that given a query, MAS-GPT first provides a reasoning process in analyzing the domain, difficulty, and required knowledge of the query. Secondly, MAS-GPT analyzes what multi-agent system is appropriate for addressing the question effectively and decides the required agents and structure.

Finally, MAS-GPT generates a query-specific MAS (Listing 1 and Figure 7(a)), which involves five agents with specific tasks targeting the query and a final decision-making agent to provide the final answer.

This indicates that MAS-GPT can flexibly generate a query-specific MAS within one single LLM inference.

## B.2. MAS-GPT Can Generalize to Unseen Queries

From Table 7, we see that given a query whose domain is unseen from our training dataset, MAS-GPT still works well. Specifically, it first analyzes the involved domain, the required understandings, and the difficulty of the given query. Secondly, it figures out a well-suited multi-agent system to answer the query, which directly mentions what agents and what structure are appropriate.

Finally, MAS-GPT generates a query-specific MAS (Listing 2 and Figure 7(b)), which involves four parallel pipelines: two pipelines that answer the query via coding and two pipelines that directly answer the query. The first two pipelines are

Table 6: Case 1: user query and the reasoning process generated by our MAS-GPT. MAS-GPT firstly analyzes the domain, difficulty, and required knowledge, then propose the idea of what multi-agent system is appropriate.

| | |
|---|---|
| Query | The following is a multiple-choice question: Which of the following statements about any two events A and B is true? 

 The following choices are provided: 
 (A) $P(A \cup B)$ implies events A and B are independent. 
 (B) $P(A \cup B) = 0$ implies events A and B are mutually exclusive. 
 (C) $P(A \cap B)$ implies events A and B are independent. 
 (D) $P(A \cup B)$ implies events A and B are mutually exclusive. 
 (E) $P(A \cap B) = 0$ implies events A and B are mutually exclusive. 
 (F) $P(A \cap B) = 1$ implies events A and B are independent. 
 (G) $P(A \cup B) = 1$ implies events A and B are independent. 
 (H) $P(A \cap B) = 1$ implies events A and B are mutually exclusive. 
 (I) $P(A \cup B) = 1$ implies events A and B are mutually exclusive. 
 (J) $P(A \cap B) = 0$ implies events A and B are independent. 

 Please finish your answer with "the answer is (X)" where X is the correct letter choice. |
| Reasoning generated by MAS-GPT | The question is a multiple-choice problem in the domain of probability theory, specifically focusing on the relationships between events, such as independence and mutual exclusivity. The difficulty lies in the need for precise understanding of probability concepts and their implications, as well as the ability to analyze and evaluate multiple statements to identify the correct one. This requires both theoretical knowledge and logical reasoning. 

 To address this question effectively, a multi-agent system should include agents that specialize in analyzing the relationships between events in probability theory. Each agent should focus on a specific aspect of the problem, such as independence, mutual exclusivity, or the implications of specific probability values (e.g., $P(A \cap B) = 0$ or $P(A \cup B) = 1$). By dividing the task into these focused analyses, the system ensures comprehensive coverage of the problem space. Additionally, a final decision-making agent is necessary to synthesize the insights from all agents and provide a well-reasoned final answer. This structure ensures that the system can handle the complexity of the question while leveraging the strengths of specialized reasoning agents. The system is designed to align with the specific requirements of the question, ensuring that each agent contributes relevant and precise insights to the solution process. |

both consisted of two agents: one for writing and executing the code, and one for organizing the final answer. Finally, an aggregation agent reasons over the four solutions and provides the final answer.

This indicates that MAS-GPT can generalize to unseen queries and generate appropriate MAS.

## B.3. MAS-GPT Can Generate Novel MAS

From Table 8 and Listing 3 (Figure 7(c)), we see that given a query, the MAS-GPT proposes a novel MAS that is unseen from our base MAS pool. Specifically, it builds an MAS with five parallel pipelines, where each pipeline actually involves an agent for answering the question and another agent for refining the solution. Finally, a decision-making agent is introduced to provide the final answer.

This shows that MAS-GPT is not merely memorizing the data during training, but actually learns patterns of designing MAS, indicating the generalization ability of our proposed MAS-GPT.

Listing 1: Case 1: Multi-agent system generated by MAS-GPT. MAS-GPT can generate query-specific MAS. MAS-GPT designs five independent responding agents, each responsible for different aspects of the task.

```
1  from utils import *
```

```python
class MAS():
    def __init__(self, model_list):
        self.llm = LLM(model_list)

    def forward(self, taskInfo):
        """
        A multi-agent system for solving probability-related multiple-choice questions.

        Steps:
            1. 5 agents analyze the task independently, each focusing on a specific
                aspect of probability theory.
            2. A final decision-making agent reasons over the solutions and provides
                the final solution.
        """

        # Step-by-step instruction for each agent to reason and generate answer
        instructions = [
            f"Task: {taskInfo}\n\nFocus on analyzing the relationship between events A
                and B in terms of independence. Solve the task.",
            f"Task: {taskInfo}\n\nFocus on analyzing the relationship between events A
                and B in terms of mutual exclusivity. Solve the task.",
            f"Task: {taskInfo}\n\nFocus on the implications of P(A \cup B) and P(A \cap
                 B) on the relationship between events A and B. Solve the task.",
            f"Task: {taskInfo}\n\nFocus on the definitions and properties of
                probability union and intersection. Solve the task.",
            f"Task: {taskInfo}\n\nFocus on identifying any logical inconsistencies or
                impossibilities in the given choices. Solve the task."
        ]

        # Call the llm to generate each solution
        cot_results = [self.llm.call_llm(instruction) for instruction in instructions]

        # Get the instruction for the final decision-making agent based on all
            generated solutions
        final_decision_instruction = self.get_final_decision_instruction(taskInfo,
            cot_results)

        # Call the llm to process the final decision-making instruction and generate
            the final answer
        final_decision_result = self.llm.call_llm(final_decision_instruction)

        # Return the final solution
        return final_decision_result

    def get_final_decision_instruction(self, taskInfo, cot_results):
        """
        Format an instruction for final decision-making based on a given task
            description and a list of solutions.

        Args:
            taskInfo (str): A description of the task that needs to be completed.
            cot_results (list): A list containing solutions or reasoning steps for the
                task.

        Returns:
            str: A formatted instruction that includes the task description, each
                solution, and a prompt for final decision-making.
        """

        # Initialize the instruction text with a general guideline
        instruction = f"Task:\n{taskInfo}\n\n"

        # Append each solution from cot_results to the instruction
        for i, result in enumerate(cot_results):
```

```
54          instruction += f"Solution {i+1}:\n{result}\n\n"  # Number each solution for
                clarity
55
56          # Add the final prompt to encourage reasoning over the solutions and provide a
              final answer
57          instruction += "Given all the above solutions, reason over them carefully and
            provide a final answer to the task."
58
59          # Return the complete instruction text
60          return instruction
```

Table 7: Case 2: user query and the reasoning process generated by our MAS-GPT. MAS-GPT firstly analyzes the domain,required elements, and the difficulty, then propose the idea of what multi-agent system is appropriate.

| Query | An electric dipole consisting of charges of magnitude $1.50$nC separated by $6.20\mu m$ is in an electric field of strength $1100$ N/C. What is the magnitude of the electric dipole moment? |
|---|---|
| Reasoning generated by MAS-GPT | The question involves calculating the magnitude of the electric dipole moment, a concept rooted in physics, specifically in the domain of electromagnetism. The problem requires understanding the relationship between the charges, their separation, and the resulting dipole moment. While the mathematical computation is straightforward, the problem demands precise application of the formula for the dipole moment and careful handling of units, which adds a layer of complexity. 

 To address this, one would analyze the question by identifying the key variables (charge magnitude, separation distance) and the formula (dipole moment = charge × separation). 

 A multi-agent system is well-suited for this task because it can leverage multiple approaches to ensure accuracy and robustness. First, agents can independently break down the problem into steps, generate Python code to compute the solution, and execute the code to verify the calculations. This ensures that the solution is not only theoretically correct but also computationally validated. Additionally, other agents can directly generate solutions without code, providing a baseline for comparison. Finally, an evaluation agent can synthesize the outputs from all agents, ensuring consistency and selecting the most reliable solution. This multi-faceted approach is particularly effective for scientific problems where both analytical and computational accuracy are critical. |

Listing 2: Case 2: Multi-agent system generated by MAS-GPT. MAS-GPT can generalize to unseen queries from SciBench (Wang et al., 2024a), generating an appropriate multi-agent system to handle the query.

```
1   from utils import *
2
3   class MAS():
4       def __init__(self, model_list):
5           self.llm = LLM(model_list)
6
7       def forward(self, taskInfo):
8       """
9       A multi-agent system for solving math problems by executing code and directly
           answering.
10      Steps:
11          1. 2 agents independently solves the problem by breaking it down into steps and
                generating code, where each agent organizes the solution based on the code
                execution results, ensuring clarity and correctness.
12          2. 2 agents generate a solution directly, which provides baseline solutions
                especially when code generation is challenging
13          3. A final agent evaluates all the solutions and determines the final solution.
14      """
15          # 4 parallel pipelines to solve the problem independently
16          solutions = []
17
```

```python
18          # The first two pipelines generate code to solve the problem
19          for _ in range(2):
20              answer, output = self.generate_code_get_output(taskInfo)
21              solution = self.organize(taskInfo, answer, output)
22              solutions.append(solution)
23
24          # The third pipeline generates a solution directly
25          for _ in range(2):
26              solution = self.llm.call_llm(taskInfo)
27              solutions.append(solution)
28
29          # Determine the final solution based on the generated solutions
30          final_solution = self.get_final_solution(taskInfo, solutions)
31          return final_solution
32
33      def generate_code_get_output(self, taskInfo):
34          """
35          Generate Python code to solve the mathematical problem and execute the code to get
              the output.
36          Args:
37              taskInfo (str): The mathematical problem to be solved.
38          Returns:
39              a tuple containing:
40                  - str: The answer generated by the LLM model.
41                  - str: The output of the code execution.
42          """
43          code_generation_instruction = f"""You are an expert in solving mathematical
              problems.
44  **Problem:**
45  {taskInfo}
46  **Instructions:**
47  1. Analyze the problem and list the steps required to solve it.
48  2. Generate Python code that can help solve the problem. The code should:
49  - Print important intermediate results in the calculation process, along with clear
      explanations.
50  - Store the final calculation result in a variable named `output`. This variable should
      contain the final result of the computation and be defined at the global scope.
51  - Be directly executable. The code should run and produce a result when executed.
52  Wrap your final code solution in <Code Solution> and </Code Solution>. For example:
53  <Code Solution>
54  Your function code here
55  </Code Solution>
56  """
57          # Call `generate_and_extract_code` to generate answer and extract the code
58          answer, code = generate_and_extract_code(llm=self.llm, prompt=
              code_generation_instruction)
59
60          # Call `execute_code` to execute the generated code and get output
61          output = execute_code(code)
62          return answer, output
63
64      def organize(self, taskInfo, answer, result):
65          """
66          Organize the solution based on the code execution results.
67          Args:
68              taskInfo (str): The mathematical problem to be solved.
69              answer (str): The initial solution generated by the LLM model.
70              result (str): The output of the code execution.
71          Returns:
72              str: The organized solution based on the code execution results.
73          """
74          organize_instruction = f"""**Problem:**
75  {taskInfo}
76  **Initial Solution:**
77  {answer}
```

```
78  **Code Execution Result:**
79  {result}
80  To solve the **Problem**, the **Initial Solution** provides steps and python code for
        calculations. The **Code Execution Result** is the output of the code.
81  Based on the **Initial Solution** and **Code Execution Result**, provide a final
        solution to the problem. Include the results of the code calculation in your
        response. Your final response should be complete as if you are directly answering
        the problem."""
82          solution = self.llm.call_llm(organize_instruction)
83          return solution
84
85      def get_final_solution(self, taskInfo, solutions):
86      """
87      Evaluate the solutions provided by the agents and determine the final solution.
88      Args:
89          taskInfo (str): The mathematical problem to be solved.
90          solutions (list): A list of solutions provided by the agents.
91      Returns:
92          str: The final solution to the mathematical problem.
93      """
94          # Prepare the solutions in a numbered format for evaluation
95          solutions_str = ""
96          for i, solution in enumerate(solutions):
97              solutions_str += f"Solution {i+1}\n{solution}\n\n"
98
99          final_solution_instruction = f"""**Problem:**
100 {taskInfo}
101 **Solutions:**
102 {solutions_str}
103 Several solutions are given to solve the problem. Carefully evaluate these solutions.
        This consistency in answers is crucial for determining the most reliable solution.
104 You should provide a final solution to the problem. Your final response should be
        complete as if you are directly answering the problem."""
105         final_solution = self.llm.call_llm(final_solution_instruction)
106         return final_solution
```

Table 8: Case 3: user query and the reasoning process generated by our MAS-GPT. MAS-GPT firstly analyzes the domain, difficulty, and required knowledge, then propose the idea of what multi-agent system is appropriate.

| Query | The following is a multiple-choice question: Suppose that there are two firms in the market facing no costs of production and a demand curve given by Q = 150 - P for their identical products. Suppose the two firms choose prices simultaneously as in the Bertrand model. Compute the prices in the nash equilibrium. |
| --- | --- |
| | The following choices are provided:
(A) 30
(B) 50
(C) 0
(D) 25
(E) 150
(F) 75
(G) 90
(H) 120
(I) 100
(J) 60 |
| | Please finish your answer with "the answer is (X)" where X is the correct letter choice. |
| Reasoning generated by MAS-GPT | The question pertains to a classic economic problem in the domain of game theory and microeconomics, specifically focusing on the Bertrand model of price competition. The difficulty lies in understanding the strategic interaction between two firms, the implications of no production costs, and the Nash equilibrium concept. Solving this requires both theoretical knowledge of economic models and the ability to apply mathematical reasoning to derive the equilibrium price. |
| | To address such a question, one must first break it down into its core components: the demand curve, the firms' identical products, and the simultaneous price-setting behavior. |
| | A multi-agent system is well-suited for this task because it can leverage multiple perspectives to ensure accuracy and depth in reasoning. By employing agents that independently analyze the problem, generate solutions, and refine their responses through iterative feedback, the system ensures a robust exploration of the problem space. This approach is particularly valuable for a question like this, where multiple solution paths (e.g., algebraic derivation, economic intuition) can lead to the correct answer. The iterative refinement process allows for cross-verification of solutions, reducing the likelihood of errors. Finally, a decision-making agent synthesizes the diverse solutions, ensuring that the final answer is both logically sound and consistent with the principles of the Bertrand model. This structured reasoning process aligns with the complexity of the question and ensures a reliable outcome. |

Listing 3: Case 3: Multi-agent system generated by MAS-GPT. MAS-GPT can generate novel MAS.

```
from utils import *

class MAS():
    def __init__(self, model_list):
        self.llm = LLM(model_list)

    def forward(self, taskInfo):
        """
        A multi-agent system for solving general tasks.
```

```
11          Steps:
12              1. 5 agents solve the task independently.
13              2. Each agent reflects on the solutions and provides an improved solution.
14              3. A final decision-making agent reasons over the improved solutions and
                   provides the final solution.
15          """
16          # Step-by-step instruction for each agent to reason and generate answer
17          instruction = f"Task: {taskInfo}\n\nPlease solve the task."
18
19          # Set the number of solutions to generate; using 5 for variety and diversity
20          N = 5
21          # Call the llm to generate each solution
22          cot_results = [self.llm.call_llm(instruction) for _ in range(N)]
23
24          # Get the instruction for the self-refine process based on all generated
                solutions
25          self_refine_instruction = self.get_self_refine_instruction(taskInfo,
                cot_results)
26
27          # Call the llm to refine each solution
28          refined_results = [self.llm.call_llm(self_refine_instruction) for _ in range(N)
                ]
29
30          # Get the final decision-making instruction based on all refined solutions
31          final_decision_instruction = self.get_final_decision_instruction(taskInfo,
                refined_results)
32
33          # Call the llm to process the final decision-making instruction and generate
                the final answer
34          final_decision_result = self.llm.call_llm(final_decision_instruction)
35
36          # Return the final solution
37          return final_decision_result
38
39      def get_self_refine_instruction(self, taskInfo, cot_results):
40          """
41          Format an instruction for self-refinement based on a given task description and
                 a list of solutions.
42
43          Args:
44              taskInfo (str): A description of the task that needs to be completed.
45              cot_results (list): A list containing solutions or reasoning steps for the
                    task.
46
47          Returns:
48              str: A formatted instruction that includes the task description, each
                    solution, and a prompt for self-refinement.
49          """
50
51          # Initialize the instruction text with a general guideline
52          instruction = f"Task:\n{taskInfo}\n\n"
53
54          # Append each solution from cot_results to the instruction
55          for i, result in enumerate(cot_results):
56              instruction += f"Solution {i+1}:\n{result}\n\n"  # Number each solution for
                     clarity
57
58          # Add the final prompt to encourage self-refinement and improvement
59          instruction += "Given all the above solutions, reason over them carefully and
                provide an improved solution to the task."
60
61          # Return the complete instruction text
62          return instruction
63
64      def get_final_decision_instruction(self, taskInfo, refined_results):
```

| METHOD | HUMANEVAL | HUMANEVAL-PLUS | MATH |
|---|---|---|---|
| CHATDEV (QIAN ET AL., 2024A) | 83.33 | 84.04 | 62.07 |
| MAS-GPT (OURS) | **91.18** | **87.23** | **77.59** |

Table 9: Comparisons with task-specific method, ChatDev (Qian et al., 2024a), which is specifically designed for software development.

```
65      """
66      Format an instruction for final decision-making based on a given task
            description and a list of refined solutions.
67
68      Args:
69          taskInfo (str): A description of the task that needs to be completed.
70          refined_results (list): A list containing refined solutions for the task.
71
72      Returns:
73          str: A formatted instruction that includes the task description, each
                refined solution, and a prompt for final decision-making.
74      """
75
76      # Initialize the instruction text with a general guideline
77      instruction = f"Task:\n{taskInfo}\n\n"
78
79      # Append each refined solution from refined_results to the instruction
80      for i, result in enumerate(refined_results):
81          instruction += f"Solution {i+1}:\n{result}\n\n"  # Number each solution for
                clarity
82
83      # Add the final prompt to encourage reasoning over the solutions and provide a
            final answer
84      instruction += "Given all the above solutions, reason over them carefully and
            provide a final answer to the task."
85
86      # Return the complete instruction text
87      return instruction
```

# C. Additional Experimental Setups.

### C.1. Descriptions of Datasets

We provide an overall descriptions of the training and testing datasets in Table 10.

### C.2. Evaluations

In this section, we detail our evaluation approach. For queries with ground truth answers, we employ LLMs to extract the MAS output and compare it with the ground truth. For code benchmarks like HumanEval and MBPP, we assess correctness using test cases.

**LLM-based Evaluation with Ground-Truth Answer** We utilize LLMs to perform evaluation with ground-truth answer. However, direct evaluation against the ground truth is incompatible as the LLM annotates the response itself. To address this issue, we adopt a two-step evaluation process based on the prompts used in AutoGen (Wu et al., 2023), first extract the answer, then evaluation. Here the responses generated by multi-agent systems (MAS) are often unstructured and irregular, making it difficult to extract the final answer to a query using rule-based methods. To avoid extraction errors that could impact the evaluation of MAS performance, we use LLMs to automate the answer extraction process. Specifically, we prompt the LLM to extract the answer from the MAS response based on predefined rules and then ask the LLM to compare it with the ground truth. The prompts used for this process are detailed in Table 13.

**Code Evaluation with Test Cases** We evaluate the MAS performance on coding tasks based on pass rate on test cases, with a two step approach: first, prompting the LLM to extract the code from the MAS response, and second, executing it in a

| Purpose | Dataset Name | Domain | Sub-Domains | Sample Number |
|---------|--------------|--------|-------------|---------------|
| Training | MATH (Hendrycks et al., 2021b) | Math | Counting & Probability
Geometry
Algebra
Number Theory
Precalculus
Prealgebra
Intermediate Algebra | 6000 |
| | GSM8K (Cobbe et al., 2021) | Math | - | 1000 |
| | GSM-Hard (Gao et al., 2023) | Math | - | 319 |
| | AQUA-RAT (Ling et al., 2017) | Reasoning | - | 1000 |
| | MBPP (Austin et al., 2021) | Code | - | 374 |
| | SciQ (Welbl et al., 2017) | General QA | - | 2000 |
| | MMLU (Hendrycks et al., 2021a) | General QA | Humanities
Social Science
STEM
Others | 1529 |
| Testing | MATH (Hendrycks et al., 2021b) | Math | Counting & Probability
Geometry
Algebra
Number Theory
Precalculus
Prealgebra
Intermediate Algebra | 500 |
| | GSM8K (Cobbe et al., 2021) | Math | - | 500 |
| | GSM-Hard (Gao et al., 2023) | Math | - | 500 |
| | HumanEval (Chen et al., 2021) | Code | - | 164 |
| | HumanEval-Plus (Liu et al., 2023) | Code | - | 164 |
| | GPQA (Rein et al., 2023) | Science | - | 448 |
| | SciBench (Wang et al., 2024a) | Science | - | 500 |
| | MMLU (Hendrycks et al., 2021a) | General QA | Humanities
Social Science
STEM
Others | 500 |
| | AIME-2024 | Math | - | 30 |

Table 10: Descriptions of benchmarks

coding environment to calculate the pass rate; see the prompts used for extract code and functions in Table 14.

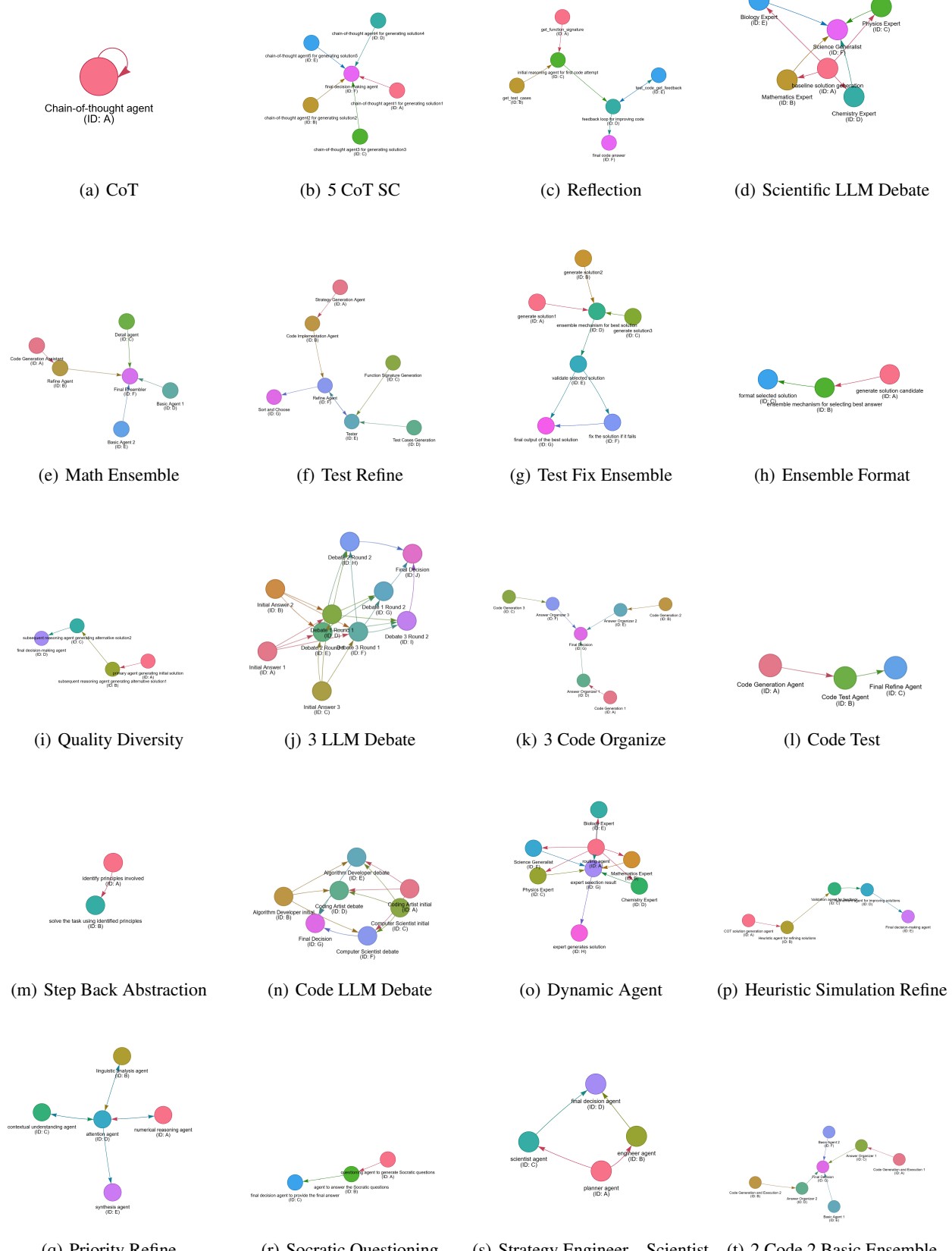

Figure 6: Visualization of our MAS pool

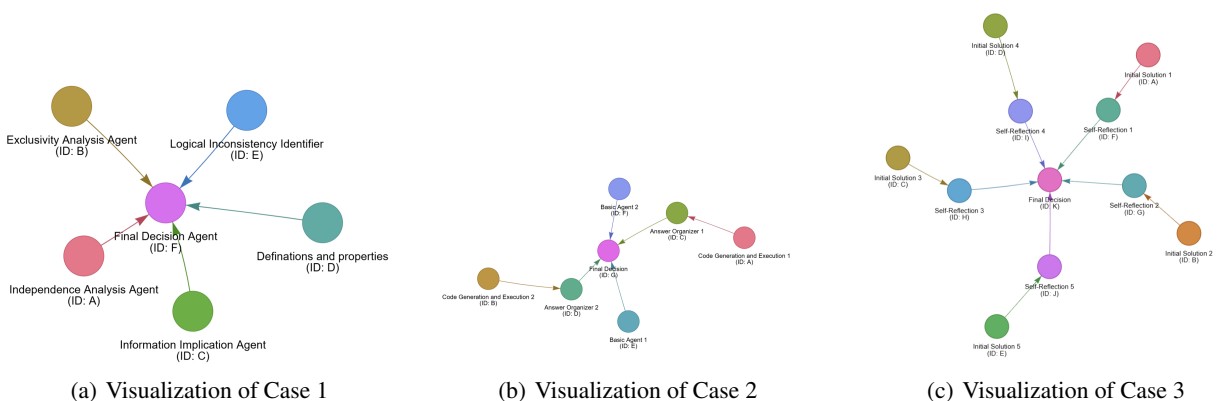

(a) Visualization of Case 1     (b) Visualization of Case 2     (c) Visualization of Case 3

Figure 7: Visualization of three cases. (a) MAS-GPT can generate query-specific MAS. (b) MAS-GPT can generalize to unseen queries. (c) MAS-GPT can generate novel MAS.

Table 11: The prompt for intra-consistency-oriented pair refinement. This prompt is fed to GPT-4o-2024-11-20 to adjust the MAS and generate a reasoning statement. The prompt is integrated with a user query and the selected MAS represented by Python code.

I will give you a question and a multi-agent system. The multi-agent system is described in the format of Python code, where each agent is represented by an agent-specific instruction and one call_llm. Though the multi-agent system can answer the question, it may not be the best one. You task is return me two things: an improved multi-agent system and a paragraph.

The improved multi-agent system should be more related to the question, while basically, try not to change the architecture compared to the original multi-agent system.
- For example, if the multi-agent system is in a parallel structure (e.g., 5 parallel agents generate answer and 1 agent select the best answer), you may keep the structure unchanged while only changing each parallel agent's instruction.
- If the multi-agent system is already suitable, you may only modify the instructions in the multi-agent system more relevant to the question while leaving the structure unchanged.
- If you think additional agents are required (e.g., the question is difficult and complex), you may add some related expert agents to enhance the multi-agent system.

The paragraph should first analyze the question itself, from the perspectives of domain and difficulty. Then, you should provide a reasoning process to bridge the question and the improved multi-agent system. The reasoning process should be in the views of that how one analyzes the question and objectively thinks about what multi-agent system is needed. Then, the reasoning process can finally and logically lead to the improved multi-agent system. Do not mention "this multi-agent system", or "the improved multi-agent system", rather, say "a multi-agent system" instead. Do not mention the original multi-agent system or the original structure.

Please follow the following format requirements:
- The improved multi-agent system should be included between `` and ``
- The paragraph should be included between `<PARAGRAPH>` and `</PARAGRAPH>`

Please firstly generate the multi-agent system and then generate the paragraph. The paragraph should analyze about the question and the generated multi-agent sytem, such that when one sees the (question, paragraph, the improved multi-agent system) triplet (wihtout the original multi-agent system), one can understand the reasoning process behind the improved multi-agent system. Notice! The paragraph should never mention the original multi-agent system or the original structure.

The question is:
`{query}`

The multi-agent system is:
`{MAS code}`

Table 12: The prompt for generating a reasoning process if the refined MAS fails. This prompt is fed to GPT-4o-2024-11-20 to generate a reasoning statement. The prompt is integrated with a user query and a selected MAS represented by Python code.

I will give you a question and a multi-agent system. The multi-agent system is described in the format of Python code, where each agent is represented by an agent-specific instruction and one call_llm. You task is return me a paragraph.

The paragraph should first analyze the question itself, from the perspectives of domain and difficulty. Then, you should provide a reasoning process to bridge the question and the provided multi-agent system. The reasoning process should be in the angle of views that how one analyzes the question and objectively thinks about what multi-agent system is needed. Then, the reasoning process can finally and logically lead to the provided multi-agent system. Do not mention "this multi-agent system", or "the provided multi-agent system", rather, say "a multi-agent system" instead.

The paragraph should analyze about the question and the provided multi-agent sytem, such that when one sees the (question, paragraph, the provided multi-agent system) triplet, one can understand the reasoning process behind the provided multi-agent system.

Remember, the paragraph should be included between ¡PARAGRAPH¿ and ¡/PARAGRAPH¿.

The question is:
**{query}**

The provided multi-agent system is:
**{MAS code}**

Table 13: Prompts for extract answer and answer evaluation.

You are a helpful AI assistant tasked with extracting the final answer from a provided solution.

**Input:**
1. A problem statement, prefixed with "===Problem: `<problem>`".
2. A solution to the problem, prefixed with "===Solution:`<solution>`".

**Problem and Solution:**
===Problem: **{query}**

===Solution: **{response}**

**Instructions:**
- Carefully analyze the solution and extract the final answer in reply: "The answer is `<answer extracted>` in reply".
- If the solution does not contain a final answer (e.g., only reasoning, code without execution, or incomplete information), respond with: "The reply doesn't contain an answer."
- Ensure that the extracted answer is exactly as presented in the solution. Do not infer or use external knowledge. Do not execute the code yourself.
- Remember, Never execute the code yourself! Never doing any computation yourself! Just extract and output the existing answer!

---

You are a helpful AI assistant. You will use your coding and language skills to verify the answer.
You are given:
1. A problem, which is going to start like "===Problem: `<problem>`".
2. A ground truth answer, which is going to start like "===Ground truth answer:".
3. A reply with the answer to the problem, which are going to start like "===Reply:".
Please do the following:
1. Extract the answer in reply: "The answer is `<answer extracted>` in reply".
2. Check whether the answer in reply matches the ground truth answer. When comparison is not obvious (for example, 3*sqrt(6) and 7.348), you may compare by calculation, allowing a small margin of error.
3. After everything is done, please give each reply a comment like the following options:
- "The answer is correct."
- "The answer is approximated but should be correct. Correct Answer: `<ground truth answer>` | Answer extracted: `<answer extracted>`."
- "The answer is incorrect. Correct Answer: `<ground truth answer>` | Answer extracted: `<answer extracted>`."
- "The reply doesn't contain an answer."
Here are the problem, the ground truth answer and the reply:
===Problem: **{query}**

===Ground truth answer: **{ground_truth_answer}**

===Reply: **{Reply}**

Table 14: Prompts for extract code and functions.

You are given a **Problem** and a **Solution**. The **Problem** asks for a code function. Extract the final code function from the **Solution**.
**Problem:**
**{query}**

**Solution:**
**{solution}**

Please follow the following rules:
- Only output the code function that exists in the **Solution**, without any additional explanation or content.
- Do not modify any part of the code function.
- Remove parts like 'example use' or 'test cases'.
- If the **Solution** does not contain a code function, respond with: "The reply doesn't contain a code function."

