# OpenReview forum: "MAS-GPT: Training LLMs to Build LLM-based Multi-Agent Systems"
_ICML.cc/2025/Conference — ICML 2025 poster_

### Official Review · Reviewer_Ap5F · 2025-03-09

**Overall Recommendation:** 4

**Summary:**

This paper introduces MAS-GPT, a novel multi-agent system generation framework. Specifically, MAS-GPT employs a multi-agent generator model trained using modules such as MAS filtering, inter-consistency assurance, and intra-consistency enhancement. Designed to adapt to the nuances of diverse tasks, MAS-GPT aims to generate optimally performing MAS for tasks of the same category, thereby enhancing the capability of LLM-MAS to address a broad spectrum of problem types. The effectiveness of MAS-GPT is rigorously validated across a wide array of benchmarks, and the paper further presents comprehensive experimental results that substantiate its capabilities.

**Claims And Evidence:**

The claims in this paper are largely consistent with the experimental results and avoid overclaiming.

**Essential References Not Discussed:**

No.

**Experimental Designs Or Analyses:**

The experimental design in this paper is largely sound and effectively demonstrates the capabilities of MAS-GPT. However, several points require further clarification and consideration:

Lack of Single Agent Framework Specification: The paper does not specify the framework employed for the single-agent baseline models.

Cost Comparisons and Metric Justification: I have reservations regarding the cost comparison metric used in this experiment. To the best of my knowledge, standard practice in evaluating LLM cost involves assessing token cost and time cost to quantify both financial and temporal overheads. Therefore, I recommend that the authors augment this section by providing token cost and time cost data for MAS-GPT and all baselines across the benchmarks used. Furthermore, the methodology for performance measurement in this cost comparison section, as well as the source of the experimental data, are not clearly delineated. Please provide this essential supplementary information.

Scaling Effects of Data Size and Supervised Fine-tuning: The "Scaling effects of data size" analysis appears inconsistent with established supervised fine-tuning methodologies. It is widely understood that fine-tuning a 32B parameter LLM with only 100 data samples is generally insufficient to achieve meaningful adaptation. Consequently, the experimental results presented in Figure 5(a) may not adequately support the conclusions drawn in this section.

**Methods And Evaluation Criteria:**

The method proposed in this paper, or more precisely, the overall pipeline for training MAS generator, effectively addresses the relevant problems and offers significant reference value.

**Other Comments Or Suggestions:**

No.

**Other Strengths And Weaknesses:**

See Experimental Designs Or Analyses.

**Questions For Authors:**

See Experimental Designs Or Analyses.

**Relation To Broader Scientific Literature:**

This paper is insightful for building LLM-based multi-agent systems.

**Theoretical Claims:**

n/a

---

> ### Author Rebuttal · Authors · 2025-04-01
>
> Thanks for your time devoted to reviewing our paper. We are glad to see that you acknowledge that our method is novel, experimental designs are sound, and our paper is insightful. It is encouraging. We would like to address your remaining concerns in the following.
>
> ---
>
> &nbsp;
>
> **Experimental Designs Or Analyses 1:** Lack of Single Agent Framework Specification: The paper does not specify the framework employed for the single-agent baseline models.
>
> **Answer:** Sorry for the missing details. We employ vLLM library to deploy LLMs. For all baseline methods, we run their official open-sourced codes.
>
> ---
>
> &nbsp;
>
> **Experimental Designs Or Analyses 2:** Cost Comparisons and Metric Justification. I have reservations regarding the cost comparison metric used in this experiment. To the best of my knowledge, standard practice in evaluating LLM cost involves assessing token cost and time cost to quantify both financial and temporal overheads. Therefore, I recommend that the authors augment this section by providing token cost and time cost data for MAS-GPT and all baselines across the benchmarks used. Furthermore, the methodology for performance measurement in this cost comparison section, as well as the source of the experimental data, are not clearly delineated. Please provide this essential supplementary information.
>
> **Answer:** Thanks for the recommendation! Following your advice, we have reported the token consumptions in addition to the number of LLM calls in the following table. In our initial experiments, we did not record token consumptions, so we need to conduct additional experiments during rebuttal. Due to limited time, we report the following results . We will report more in our revision. From the table, we see that our MAS-GPT achieves the best performance with the least inference costs (both call times and token consumptions).
>
> The performance measurement uses accuracy as metric, which is averaged on all benchmarks in Table 2. And the specific numbers are taken from Table 2. We will include all of these in our revision.
>
> |              | AgentVerse    | DyLAN         | MAS-GPT                       |
> | ------------ | ------------- | ------------- | ----------------------------- |
> | LLM Calls    | 12.05 (70B)   | 12.96 (70B)   | 1 (32B) + 6.44 (70B)          |
> | Tokens       | 8610.07 (70B) | 4874.22 (70B) | 1133.18 (32B) + 2126.98 (70B) |
> | Accuracy (%) | 59.36         | 60.54         | 64.47                         |
>
>
> ---
>
> &nbsp;
>
> **Experimental Designs Or Analyses 3:** Scaling Effects of Data Size and Supervised Fine-tuning: The "Scaling effects of data size" analysis appears inconsistent with established supervised fine-tuning methodologies. It is widely understood that fine-tuning a 32B parameter LLM with only 100 data samples is generally insufficient to achieve meaningful adaptation. Consequently, the experimental results presented in Figure 5(a) may not adequately support the conclusions drawn in this section.
>
> **Answer:** Thanks for this valuable comments. We would like to answer from two perspectives.
>
> (1) The base model here is a 32B-Instruct model, which has coding capability at some levels. Therefore, training on 32B-Instruct models to learn to generate MAS code may have difference compared to training 32B-pretrained models to learn instruction-following capabilities.
>
> (2) Despite that Figure 5 (a) shows that using 100 training samples could significantly reduce the error rate of execution, it does NOT indicate that using 100 samples is sufficient. Please note that in Figure 5 (b), using 100 training samples results in significantly lower perfomance compared to using 10000 samples. That is, using 100 samples may successfully teach the LLM to generate executable MAS code, but fails to teach it to generate appropriate MAS, which is essential in ensuring good performance.
>
> We truly believe in this new paradigm. With more diverse and high-performance MAS being included, we believe that MAS-GPT will be further advanced in a way similar to the advancement of ChatGPT: with better data and training techniques, the models become better.
>
> ---
>
> &nbsp;
>
> Overall, we hope that our responses can fully address your concerns and will be grateful for any feedback.

---

### Official Review · Reviewer_xR52 · 2025-03-09

**Overall Recommendation:** 2

**Summary:**

This paper proposes to train an LLM to build multi-agent system. The paper reframes MAS construction as a python coding task and represents MAS as executable python code. One key contribution is a consistency-oriented data construction pipeline that generates high-quality query-MAS pairs. Extensive experiments on different benchmarks and backbones indicate that MAS-GPT outperforms baselines in effectiveness, efficiency and generalization capability. MAS-GPT also excels in terms of inference costs.

**Claims And Evidence:**

Yes.

**Essential References Not Discussed:**

Works that involve optimizing MAS architecture:
[1] Cut the Crap: An Economical Communication Pipeline for LLM-based Multi-Agent Systems. ICLR 2024

**Experimental Designs Or Analyses:**

Yes.

**Methods And Evaluation Criteria:**

Yes.

**Other Comments Or Suggestions:**

N/A

**Other Strengths And Weaknesses:**

Strengths:
1. The paper presents a novel idea by training an LLM to build multi-agent systems, thereby reducing both inference cost and the manual effort involved in system design. The motivation is clearly articulated, positioning the work as a natural evolution in the use of multi-agent orchestration.
2. The authors provide a comprehensive description of their consistency-oriented data construction pipeline. The detailed rational behind the strategies enhances the credibility of the proposed pipeline.
3. The presentation quality is good. The paper writeup is clear and easy to follow.

Weaknesses:
1. Representing multi-agent system as python code neglects important aspects of agent functionality, such as tool integration and multi-turn interactions. In practical agent systems, agents interact with their environments by executing tools and processing iterative feedback. However, in the scope of this research, the agent seems to be prompting LLM in a single-turn manner. Therefore, the approach somewhat resembles automated workflow orchestration or bootstrapped reasoning. This oversimplification is my main concern of this paper.
2. The paper should add discussion and compare the performance and cost with AgentPrune[1], a pipeline for multi-agent communication.
3. The method shares similarities with ADAS in its representation of MAS as code. The paper would benefit from a direct performance and cost comparison with ADAS, similar to the comparison provided with AFlow.
4. The performance gain seems modest. In table 2, when compared to single LLM, MAS-GPT shows modest performance gains on most benchmarks except MATH and GSM-H. In table 3, similar observation is there for qwen2.5b model. This raises questions about the practical benefits of integrating multi-agent structures, especially given the additional complexity involved.

[1]  Cut the Crap: An Economical Communication Pipeline for LLM-based Multi-Agent Systems. ICLR 2024

**Questions For Authors:**

1. For the generated MAS, have you observed any interesting workflows beyond the typical sequential prompting and answer ensembling? For example, are there cases where the MAS incorporates if-else structures, cross-references answers from different agents, or employs other non-linear strategies in its workflow?
2. In Figure 7(l), you include a "code test agent." Could you explain how this agent works in practice? Does it verify the generated code by actually executing it with test cases, or does it simply deduct whether the output aligns with the expected test case in natural language?
3. The data curation process involves multiple selection and refinement steps to construct the final query-MAS pair dataset. Could you provide details on the number of query-MAS pairs retained after each selection step (e.g., after initial pairing, after inter-consistency selection, and following intra-consistency refinement)?

**Relation To Broader Scientific Literature:**

The paper builds on and extends research in automatic and adaptive construction of agentic systems. In contrast to previous works that require manual design or multi-round refinement, this paper proposes training an LLM to generate MAS in one inference, thereby reducing cost and improving efficiency. This direction is aligned with a broader trend toward developing more streamlined, scalable, and adaptive AI architectures that simplify complex system design through end-to-end learning.

**Theoretical Claims:**

N/A. No theoretical claims involved.

---

> ### Author Rebuttal · Authors · 2025-04-01
>
> We are glad that you acknowledge that our idea is novel, method is comprehensive, and direction is aligned with a broader trend toward developing more streamlined, scalable, and adaptive architectures. Let's address your remaining concerns!
>
>
> ---
>
> **W1:** Representing MAS as python code neglects important aspects of agent functionality...?
>
> **A:** Sorry for the missing details! Our MAS-GPT actually **supports** functionalities mentioned by you (tool and multi-turn interactions) and they are indeed compatible with code.
>
> (1) Code execution tool is implemented as a python function `execute_code(code_text)`, which outputs the execution results to other agents. More tools can be implemented similarly.
>
> (2) While single-turn LLM call is represented by a function `call_llm(prompt)`, multi-turn interaction is represented by `multi_turn(history, prompt)`.
>
> Representing MAS as python code is a promising direction, as most functionalities in our AI community can be represented by python code. Along this new direction, we believe that with more data, tools, MAS, MAS-GPT could be continuingly improved as that from GPT-3.5 to GPT-4.
>
> ---
>
> **W2:** Discussion on AgentPrune.
>
> **A:** Thanks for recommendation! We will include these discussions and cite it.
>
> (Methodology) The goals of our paper and AgentPrune are significantly different. We aim to optimize MAS-GPT **once** on diverse domains such that the MAS-GPT can **generalize** to diverse domains. While AgentPrune aims to optimize on one validation dataset such that it can work on the corresponding test dataset. There are also two recent papers with goals similar to AgentPrune: GPTSwarm(ICML2024 oral), AFlow(ICLR2025 oral). While these three require **re-optimizing** for each test set, MAS-GPT can generalize to diverse test sets **without re-optimizing**.
> Since in practice, users will not provide several examples in advance for optimization, we believe that MAS-GPT is a promising direction to truly make MAS practically applicable.
>
> (Experiments) We run the official code of AgentPrune on HumanEval and MMLU. From the table, we see that our method performs **much better with less inference cost.**
>
> ||Method|Acc|Re-optimizing cost|Test cost (avg)|
> |-|-|-|-|-|
> |HEval|AgentPrune|70.12|349(70B)|9.8(70B)|
> |HEval|MAS-GPT|80.25|0|1(32B)+8.7(70B)|
> |MMLU|AgentPrune|75.05|280|7.3(70B)|
> |MMLU|MAS-GPT|78.38|0|1(32B)+4.7(70B)|
>
> ---
>
> **W3:** Comparison with ADAS.
>
> **A:** Thanks for the advice.
>
> (1) The reason why we compare with AFlow rather than ADAS is that AFlow is an improved follow-up of ADAS.
>
> (2) We compare MAS-GPT with ADAS (optimized on GPQA). MAS-GPT performs **significantly better than ADAS on these diverse datasets at a much lower cost**, indicating that MAS-GPT is much more generalizable.
>
> ||MATH|GSM8K|GSM-H|H-Eval|H-Eval+|MMLU|GPQA|SciBench|Avg|LLM calls|
> |-|-|-|-|-|-|-|-|-|-|-|
> |ADAS|34.7|36.3|12.4|75.9|69.6|76.6|38.1|14.5|44.8|21~41 (70B)|
> |MAS-GPT|68.7|93.4|62.4|80.3|78.9|78.4|37.6|24.2|65.5|1 (32B) + 6.4 (70B)|
> ---
>
> **W4:** The performance gain seems modest?
>
> **A:** Thanks for the comments. We would like to respond from the following perspectives.
>
> (1) MAS-GPT is the **only method** that consistently achieves better performance than SINGLE. Here, we report the percentage of benchmarks where a method outperforms a single agent.
>
> ||CoT|SC|AgentVerse|DyLAN|MAS-GPT|
> |-|-|-|-|-|-|
> |Percentage|50|62.5|37.5|50|100|
>
> (2) On challenging datasets such as AIME-2024 and GSM-H, our method achieves significantly better performance than SINGLE (16.67 on AIME-2024), demonstrating substantial potential.
>
> (3) We agree that for some benchmarks, improvements are not significant. Please note that these patterns commonly exist in all methods. The reasons why we still report them are twofold. First, these benchmarks are commonly used in MAS literature and we follow their setups. Second, we want to show that our method is generalizable on diverse datasets.
>
> ---
>
> **Q1:** Are there cases where the MAS incorporates if-else structures...?
>
> **A:** Yes, we have observed interesting workflows. For example, we see MAS with if-else structures in several cases. (1) Agents evaluate the current status and determine whether to stop or continue solving. (2) If code executor shows that the code runs correctly, then the system ends; else the system continues.
>
> ---
>
> **Q2:** "code test agent"?
>
> **A:** Yes. We verify the generated code by actually executing it. We implement two related python functions: one takes code piece as input and one takes code piece together with test cases as inputs.
>
> ---
> **Q3:** Number of query-MAS pairs?
> **A:** Thanks for the advice. The selection process will filter out those queries that all MAS fail to correctly answer. The refinement process essentially replace those original query-MAS pairs(Line 256-258).
>
> |Step|Init|Select|Refine|
> |-|-|-|-|
> |Num|12292|11442|11442|
>
> ---
>
> Overall, we hope that our responses can fully address your concerns and will be grateful for any feedback.

---

> > ### Comment · Reviewer_xR52 · 2025-04-09
> >
> > I appreciate the authors' detailed response, but I still have significant reservations regarding several aspects:
> >
> > 1. Regarding Weakness 1. My point is that the examples provided in the current paper do not fully capture the complexity inherent in realistic multi-agent systems. I acknowledge your clarification that the representation of MAS as Python code can theoretically support tools and multi-turn interactions. However, the current implementation, as demonstrated in the appendix, only uses the function `self.llm.call_llm(instruction)`, which indicates a single-turn LLM interaction. This does not incorporate the `multi_turn` function or other advanced capabilities mentioned in your response. Therefore, my original concern remains: the MAS instances generated and trained by MAS-GPT appear overly simplistic, essentially serving as ensembles or variants of multi-agent debate, rather than sophisticated, interactive multi-agent systems.
> >
> > 2. Regarding Performance. I acknowledge your argument that MAS-GPT demonstrates consistent performance improvements across benchmarks. However, it's important to note that the AIME dataset contains only 30 tasks. Therefore, a seemingly substantial improvement of 16% means only 5 additional correct answers. This absolute gain limits the practical significance of the improvement, especially given the complexity introduced by the multi-agent structure.
> >
> > 3. I am not fully convinced about your explanation regarding the "code test agent". How exactly does your current implementation incorporate an agent with code execution capabilities if your agent is called by `self.llm.call_llm()`? Additional details on how these specialized agents are integrated into your code is needed.
> >
> > 4. Regarding your `utils` file. I also have questions about the `utils` module imported in your code examples. It is not clear what auxiliary tools and functions are included in `utils`. Additionally, since the generated code consistently begins with `from utils import *`, can you clarify how MAS-GPT learns about the complete list of the utility functions? How do you ensure MAS-GPT can avoid generating or referencing utility functions that do not exist?

---

> > > ### Author Response · Authors · 2025-04-09
> > >
> > > Thanks for the reply. We noticed that most of your reservations are caused by insufficient details about implementation. Please allow us to provide all details you are interested.
> > >
> > > ---
> > >
> > > **Q1:** The current implementation, as demonstrated in the appendix, only uses the function self.llm.call_llm(instruction)?
> > >
> > > **A1:** We kindly remind the reviewer that the current implementation **already supports** the tools mentioned is our reponse to **W1** (i.e., execute_code, call_llm, multi_turn). Please refer to evidence in **Line 841-846** in appendix. In this example, agent first generates the code by calling `generate_and_extract_code` and then executes the code by calling `execute_code`. These two functions are implemented in the `from utils import *` (details provided in response to *Q3&Q4*).
> > >
> > > We are sorry that we did not emphasize this detail and will make this clear in our revision. We will open source all codes, data, and models.
> > >
> > > &nbsp;
> > >
> > > ---
> > >
> > > **Q2:** Regarding Performance.
> > >
> > > **A2:**
> > >
> > > (1) We agree that AIME has limited sample amount. During rebuttal, we experimented on GAIA dataset, where we see that MAS-GPT outperforms singe agent with a significant margin.
> > >
> > > (2) Meanwhile, we believe it is crucial to view this from a comparative perspective. We conduct a thorough comparison across many baselines (10 + 2 in rebuttal), all of which were implemented using official code. Please note that GPTSwarm (ICML 2024 oral) compares with 4 baselines while AFlow compares with 7 baselines. From this comparative standpoint, our method outperforms existing approaches, demonstrating its effectiveness. While we could have chosen to highlight only the benchmarks with larger improvements, we intentionally included a broader range of results to provide the community with a more comprehensive view.
> > >
> > > (Qwen2.5-72B-Instruct, samples without additional files)
> > > ||L1|L2|
> > > |-|-|-|
> > > |Single|16.67|9.23|
> > > |MAS-GPT|23.81|21.54|
> > >
> > > &nbsp;
> > >
> > > ---
> > >
> > > **Q3:** “Code agent”implementation?
> > >
> > > **A3:** Sorry for that our verbal descriptions did not provide you a clear understanding. Please allow us to straightforwardly show you the implementation.
> > > ```
> > > def execute_code(code):
> > >     if not code:
> > >         return "Empty code. No output."
> > >
> > >     temp_dir = tempfile.mkdtemp()
> > >
> > >     output_dict = {"output": None, "stdout": None, "error": None}
> > >
> > >     def run_code():
> > >         try:
> > >             global_vars = {}
> > >             local_vars = {}
> > >
> > >             # Write the code to a temporary file
> > >             with open(os.path.join(temp_dir, "script.py"), "w", encoding="utf-8") as f:
> > >                 f.write(code)
> > >
> > >             # Capture standard output
> > >             stdout_capture = io.StringIO()
> > >             with contextlib.redirect_stdout(stdout_capture):
> > >                 exec(code, global_vars, local_vars)  # Execute code
> > >
> > >             output_dict["stdout"] = stdout_capture.getvalue().strip()  # Capture print() output
> > >             output_dict["output"] = local_vars.get("output", "None")  # 'output' variable
> > >
> > >         except Exception as e:
> > >             output_dict["error"] = traceback.format_exc()
> > >
> > >     if output_dict["error"]:
> > >         return f"Error:\n{output_dict['error']}"
> > >
> > >     return f"Final output: {output_dict['output']}\nPrint during execution:\n{output_dict['stdout']}"
> > > ```
> > >
> > > &nbsp;
> > >
> > > ---
> > >
> > > **Q4:** utils file.
> > >
> > > **A4:** Sorry for the confusion. Totally, we implement the following class and functions in `utils`, which are described in MAS-GPT's system prompt :
> > > ```
> > > - `LLM(model_list)`: a class that represents an LLM with the given model list, with two available functions: call_llm(self, prompt) and multi_turn(self, history, prompt).
> > > - `execute_code(code)`: a function that executes the given code and returns the output.
> > > - `test_code_get_feedback(code, test_cases)`: a function that tests the given code with the test cases and returns the feedback.
> > > - `get_function_signature(llm, taskInfo)`: a function that returns the generated function signature for the given task.
> > > - `get_test_cases(llm, taskInfo, function_signature)`: a function that returns the generated test cases for the given task and function signature.
> > > - `extract_code_solution(solution)`: a function that returns the code by extracting (wrapped within <Code Solution> and </Code Solution>) from the given solution.
> > > - `generate_and_extract_code(llm, prompt, temperature=None)`: a function that returns the generated response and the extracted code from the response.
> > > ```
> > >
> > > Due to limited space, please refer to the example of implementing `execute_code` in **A3**.
> > >
> > > Since the training data include many examples of using functions that are provided in system prompt and examples where some new functions are implemented in the code representing MAS (see example in Line 732-757), MAS-GPT can learn to use available functions in the system prompts and also self-implement python functions for usage.
> > >
> > > ---
> > >
> > > Thanks for mentioning these. We will open source all codes, data, and models. Looking forward to your feedback and re-evaluation!

---

### Official Review · Reviewer_w8Qs · 2025-03-11

**Overall Recommendation:** 2

**Summary:**

The paper introduces MAS-GPT, a novel approach that trains LLMs to automatically generate query-specific multi-agent systems (MAS) in a single inference step. Unlike previous methods requiring manual configuration or multiple LLM inferences, MAS-GPT simplifies MAS creation by reframing it as a generative language task, where the MAS is output as executable Python code tailored to user queries. The authors propose a consistency-oriented dataset construction pipeline to produce high-quality training data, enabling MAS-GPT to effectively learn to build adaptive MAS. Experiments on nine diverse benchmarks show that MAS-GPT consistently outperforms existing multi-agent methods, achieving better adaptability, generalization, and significantly lower computational costs.

**Claims And Evidence:**

The primary claim of this paper is that MAS-GPT significantly improves adaptability, generalization, and computational efficiency compared to existing methods. To substantiate this claim, the authors conducted experiments across nine datasets.

However, I find that the experimental setting presented in this paper is toy and not real, as the training and testing data are derived from the same domain. If the authors wish to convincingly demonstrate MAS-GPT’s generalization capability, they should conduct additional experiments on out-of-domain datasets, such as GAIA.

Additionally, the authors argue that MAS-GPT offers improved computational efficiency over manually designed multi-agent systems. Nevertheless, their evaluation does not account for the additional computational costs incurred by model training and dataset refinement, both of which likely require significant time investment. In the authors setting, the users need to train "MAS-GPT" every time for each domain separately.

**Essential References Not Discussed:**

N/A

**Experimental Designs Or Analyses:**

I checked all experiment results and settings. The critical issue is what I mentioned in previous part. The training and test data are from the same domain. I greatly suspect the usability of this method.

**Methods And Evaluation Criteria:**

The authors use the metrics from the benchmarks they use in experimental section. The metrics are good.

**Other Comments Or Suggestions:**

The authors should further elaborate in their paper—particularly through experiments or additional analysis—why MAS-GPT is more efficient compared to manual MAS design.

**Other Strengths And Weaknesses:**

**Strengths**

1. The paper is well-written, I could easy follow the reasoning logic. Overall, it is easy to understand.

2. I agree the motivation of this paper. Building MAS manually is not practical. Thinking of the application/query grows, the human effort needed will be large.

**Weakness**

1. The experimental setup appears simplistic and unreliable. The authors trained MAS-GPT and evaluated it within the same domain, suggesting that users would need to retrain MAS-GPT every time they apply it to a new domain. When considering the training time required for MAS-GPT, I doubt it would actually be more efficient than manual MAS design.

2. In Table 2, the performance difference between "Single" and proposed method is not obvious. Most the performance improvement seems coming from the GSM-H dataset. Improvements on other datasets are small and seems randomness. Multiple evaluations runss needed to be conducted.

3.  The authors spent considerable effort manually refining MAS within their training dataset. As a result, I suspect that the observed performance improvements largely originate from the manually designed MAS, and that MAS-GPT essentially memorizes these manually provided MAS inputs.

**Questions For Authors:**

1. In Table 2, how many experimental runs did the authors perform? Could the authors explain why the differences across datasets are relatively small?

2. Why did the authors train and test MAS-GPT within the same domain? When claiming that MAS-GPT is efficient, did the authors account for the development and training time of MAS-GPT itself?

**Relation To Broader Scientific Literature:**

N/A

**Theoretical Claims:**

There is no theoretical claims involved in the paper.

---

> ### Author Rebuttal · Authors · 2025-04-01
>
> We are glad to see that you agree with our motivation and that our approach is novel. We notice that your main concern is on the experimental setups, of which we believe is caused by some confusion. (Since the comments in `Claims` are overlapped with `Weaknesses`, so we focus on the latter)
>
> ---
>
> &nbsp;
>
> **W1:** The authors trained MAS-GPT and evaluated it within the same domain?
>
> **Answer:** Sorry for the confusion.
>
> (1) We would like to highlight that we do **NOT** 'train MAS-GPT and evaluate it within the same domain'. In contrast, we train MAS-GPT on diverse domains simultaneously and evaluate them on diverse domains **without re-training**. For example, during evaluation, GPQA (graduate-level QA), SciBench (college-level scientific problems), AIME-2024 (mathematical competition) are all **out-of-domain benchmarks and much harder** than the training data.
>
> (2) Following your advice, we report performance results in the medical domain, MedQA (there is no medical dataset in training). From the table, we see that MAS-GPT is indeed generalizable.
>
> ||AgentVerse|DyLAN|MAS-GPT|
> |-|-|-|-|
> |MedQA|65.84|76.34|78.60|
>
> (3) Compared to many optimization-based methods, MAS-GPT is indeed more generalizable. GPTSwarm (ICML 2024 oral), AFlow (ICLR 2025 oral), and AgentPrune (ICLR 2025) are all benchmark-dependent methods. That is, to test their methods on one benchmark, they need to firstly optimize on a subset (with GT labels) of that benchmark. In contrast, our MAS-GPT is optimized on diverse training data and then **generalize to many benchmarks without modification**.
>
> ---
>
> &nbsp;
>
> **W2:** The performance difference between "Single" and proposed method is not obvious?
>
> **Answer:** Sorry for the missing details. We run twice. We would like to emphasize from three perspectives.
>
> (1) MAS-GPT is the only method that consistently achieves better performance than SINGLE. Please refer to the following table, where we report the percentage of benchmarks where a method outperforms a single agent.
>
> |Method|CoT|SC|AgentVerse|DyLAN|MAS-GPT|
> |-|-|-|-|-|-|
> |Percentage|50|62.5|37.5|50|100|
>
> (2) On particularly challenging datasets such as AIME-2024 and GSM-H, our method achieves significantly better performance than SINGLE (e.g., 16.67 on AIME-2024), demonstrating substantial potential.
>
> (3) We agree that for some benchmarks, improvements are not significant. Please note that these patterns commonly exist in all methods. The reasons why we still report them are twofold. First, these benchmarks are commonly used in MAS literature and we follow their setups. Second, we want to show that our method is generalizable on diverse datasets.
>
> ---
>
> &nbsp;
>
> **W3:** Improvements originate from manually designed MAS?
>
> **Answer:** Thanks for the comments. We would like to address your concerns from two perspectives.
>
> (1) Following your advice, we compute the number of unique MAS during test time (GSM-Hard, MATH, and SciBench) compared to those in training data. This strongly verify that MAS-GPT does **NOT** essentially memorize the training data.
>
> ||GSM-Hard|MATH|SciBench|
> |-|-|-|-|
> |Unique/total|925/1000|690/1000|491/692|
>
> (2) It is NOT a bad thing if MAS-GPT sometimes generates MAS that exists in the training data. The key of training MAS-GPT is making it learn to generate appropriate MAS based on each specific query. The keyword here is `appropriate` but not `new`. This is actually similar to the training of LLMs, LLMs could generate identical sentences seen during training and also new sentences as long as they are appropriate.
>
> ---
>
> &nbsp;
>
> **Q1:** How many runs?
>
> **Answer:** We performed 2 runs. See W2.
>
> ---
>
> &nbsp;
>
> **Q2:** Why did the authors train and test MAS-GPT within the same domain? When claiming that MAS-GPT is efficient, did the authors account for the development and training time of MAS-GPT itself?
>
> **Answer:** We did NOT train and test MAS-GPT within the same domain. Please refer to responses in Weakness 1.
>
> Please allow us to emphasize our focus on this paper. Our ultimate goal is to make MAS-itegrated application effective and efficient during inference. To achieve this, we consider a brand-new paradigm: training an LLM. This LLM (i.e., MAS-GPT) is trained on diverse data and can generalize to diverse scenarios, making it a step closer to this goal. Training is **one-time while inference could be endless** (just like OpenAI trains GPT-4 for one time and serves the world countless times). Training-time efficiency is not the focus of this paper, which could be a potential future direction.
>
> As an analogy, in efficient LLM research, their goal is to ensure the trained model enables LLMs to be efficient in applications (inference), rather than focusing on training efficiency. Our work shares a similar objective.
>
> ---
>
> &nbsp;
>
> Overall, we hope that our responses can fully address your concerns and will be grateful for any feedback.

---

> > ### Comment · Reviewer_w8Qs · 2025-04-07
> >
> > Thank you for your response. My concerns are still there:
> >
> > [1] W1: As I previously comment, the majority of the performance gains appear to come from the Math domain. This largely due to the training set includes Mathmatic problems. Similarly, SciQ (training) and SciBench (testing) are from the same domain. Regarding MedQA, It's not truly out-of-domain. You include large general QA problems in the training set. As I suggested before, GAIA is one example that could truly represents an out-of-domain setting.
> >
> > [2] You state: "It is NOT a bad thing if MAS-GPT sometimes generates MAS that exists in the training data." — **there is no doubt it is a bad thing**. Your motivation is to eliminate the need for manual MAS design, yet this implies that humans must still manually curate MAS examples for training (even more cost). That seems to contradict your original motivation. From the table you listed,  the number of unique MAS greatly support my assumptions (except GSM-Hard), MAS-GPT memory the MAS in the training set. Additionally, why not show these information on all test data but just cherrypick three?
> >
> > [3] it is difficult for me to accept the explanation that you **had already** run the experiments twice but **forgot** to report the details in the paper.
> >
> > [4] The clarification regarding the performance comparison with the single-agent baseline makes sense to me.
> >
> > I was going to lower your score because you avoided some facts, but considering that you clearly explained your advantages over single agents, I will maintain the original score.

---

> > > ### Author Response · Authors · 2025-04-07
> > >
> > > Thanks for the reply. There are some misunderstandings and we would like to further address your concerns.
> > >
> > > &nbsp;
> > >
> > > ## Contexts
> > > Firstly, please allow us to introduce the progress of the MAS research to ensure that our contexts are well aligned. There are broadly three types of MAS works:
> > >
> > > - **Type 1:** Manual design. Manually designed for some specific tasks (such as coding): MetaGPT [1], ChatDev [2].
> > > - **Type 2:** Test-time optimization. Rely on LLMs with multiple LLM calls to optimize the MAS and then solve the query: DyLAN [3].
> > > - **Type 3:** Validation-set-required optimization. Optimize the MAS on a subset from test dataset (e.g., MATH). Then, the MAS is tested on the corresponding test dataset (also MATH). That is, the training and testing datasets are exactly from the **same source**! Examples include GPTSwarm [4, ICML 2024 Oral], ADAS [5, ICLR 2025], AFlow [6, ICLR 2025 Oral], AgentPrune [7, ICLR 2025]. We have now compared with all of these methods!
> > >
> > > However, considering applying MAS in practice (e.g., serve the world like ChatGPT), they all fail:
> > > - **Type 1 -> Inadapativity:** The real-world user queries are diverse, where the fixed manually-designed MAS would fail.
> > > - **Type 2 -> Cost-inefficiency:** Optimizing the MAS for each query with many LLM inference is cost-intensive for wide application.
> > > - **Type 3 -> Lack of generalization:** In [4,5,6,7], every time they switch to another test dataset, their MAS need to be **re-optimized** (cost hundreds or thounds LLM calls). However, user queries are diverse and there is no related examples available in advance, making them inapplicable.
> > >
> > > Addressing these, MAS-GPT offer three key advantages:
> > > - **Adaptivity.** MAS-GPT adaptively generates suitable query-specific MAS.
> > > - **Cost-efficiency.** Building MAS requires only ONE inference of a 32B-sized model rather than multiple calls of strong models like GPT-4o.
> > > - **Generalization.** After training only **ONCE**, MAS-GPT generalizes to many unseen domains and significantly more challenging tasks without retraining.
> > >
> > > &nbsp;
> > >
> > > ## Answers
> > > **C1:** As I previously…
> > >
> > > **A1:**
> > > (1) Though you thought that MAS-GPT is not generalizable enough, please note that **MAS-GPT is the most generalizable method currently!** Rather than optimizing on one subset of the same source as test set before testing [4,5,6,7] (i.e., generalize to only **1 same-source** test set for each optimization), MAS-GPT can generalize to **4 same-source test datasets and 5 different-source** test datasets for ONE optimization! This should not be overlooked.
> > >
> > > (2) Why we did not try GAIA: we misunderstood your point. We thought the concerns result from that we did not provide sufficient descriptions of test datasets. Meanwhile, we believe that compared to other works, our test datasets can better verify generalization (different sources, more difficult).
> > >
> > > (3) We are now working our best on GAIA, please give us some time! We will include these in the revision. Thanks for recommendation.
> > >
> > > (Qwen72b, samples without extra files, no tool provided) **MAS-GPT generalizes to GAIA!**
> > > ||L1|L2|
> > > |-|-|-|
> > > |Single|16.67|9.23|
> > > |SC|19.05|13.85|
> > > |MAS-GPT|23.81|21.54|
> > >
> > > ---
> > >
> > > **C2:** You state:..
> > >
> > > **A2:**
> > >
> > > (1) This does NOT contradict our motivation and there are misunderstandings. Our key motivation is to **generate appropriate MAS given any query in practice** (serving like ChatGPT). Our claim is that manually-designed MAS fail to apply in such scenarios while MAS-GPT works by `standing on the shoulders of giants`! (an exciting property). Ideally, if we include all existing MAS methods (**either manually designed or optimized by LLMs**) into training, in practice, deploying one MAS-GPT can solve diverse user queries efficiently!
> > >
> > > (2) Why we report these three: in Table 2, MATH, GSM-H, and SciBench are three benchmarks where MAS-GPT has the most improvement. They best reflect that our gain does not stem from pure memorization.
> > >
> > > ---
> > >
> > > **C3:** it is difficult…
> > >
> > > **A3:** We are wrongfully accused. (1) We planned to run three times. However, due to time and budget constraints, we were only able to run twice. At the time, we did not consider it significant enough to emphasize. (2) We reported results on many benchmarks and baselines, which also make our results convincing. (3) We will open source all codes, data, and model.
> > >
> > > ---
> > >
> > > We sincerely hope that you will consider our article from the perspective of the current progress in the MAS research community and hope you could re-evaluate our paper. Thanks!
> > >
> > > [1] Metagpt: Meta programming for multi-agent collaborative framework, ICLR 2024 Oral
> > >
> > > [2] Chatdev: Communicative agents for software development, ACL 2024
> > >
> > > [3] A dynamic LLM-powered agent network for task-oriented agent collaboration, COLM
> > >
> > > [4] Language Agents as Optimizable Graphs, ICML 2024 Oral
> > >
> > > [5] AUTOMATED DESIGN OF AGENTIC SYSTEMS, ICLR 2025
> > >
> > > [6] AFLOW: AUTOMATING AGENTIC WORKFLOW GENERATION, ICLR 2025 Oral
> > >
> > > [7] CUT THE CRAP: AN ECONOMICAL COMMUNICATION ..., ICLR 2025

---

### Official Review · Reviewer_2xyd · 2025-03-14

**Overall Recommendation:** 2

**Summary:**

The paper presents MAS-GPT, a novel approach that automates the creation of multi-agent systems specifically tailored to user queries using a single inference. The authors address key limitations in existing MAS approaches, namely high manual crafting effort and high computational costs, and propose to simplify MAS construction as a generative language task. They introduce a dataset construction pipeline emphasizing consistency, which facilitates supervised fine-tuning of MAS-GPT. Extensive experiments demonstrate MAS-GPT’s superiority across various tasks, proving its effectiveness, efficiency, and adaptability.

**Claims And Evidence:**

Yes

**Essential References Not Discussed:**

Yes

**Experimental Designs Or Analyses:**

Yes

**Methods And Evaluation Criteria:**

Yes

**Other Comments Or Suggestions:**

Refer to the Questions.

**Other Strengths And Weaknesses:**

**Strengths**:
1. The inter- and intra-consistency-oriented approach is robust, methodologically sound.
2. Thorough experiments with diverse benchmarks and backbones validate the generality and effectiveness of MAS-GPT.
3. The writing is easy to follow.

**Weaknesses**:
1. The generalization capability of MAS-GPT across significantly different or novel task domains remains unclear. Although the authors designate benchmarks such as H-Eval and SciBench as out-of-domain, their training dataset explicitly includes MBPP (programming benchmark similar to H-Eval) and SciQ (science question-answering dataset similar to SciBench). Consequently, these tasks are not strictly out-of-domain. The LLM could memorize task and MAS patterns during sft and subsequently reproduce them during inference.
2. The generated MAS topologies presented are relatively straightforward. This simplicity raises concerns about MAS-GPT’s potential to effectively handle complex real-world tasks that require sophisticated interactions and multi-agent collaboration.

**Questions For Authors:**

Questions:
1. In Table 1, the number of MAS is 7580. What is your criteria for distinguishing two MAS instances? Specifically, if two MAS share an identical topology but differ in the instructions to the agent, are they counted as separate MAS instances, or are they considered identical?
2. How does MAS-GPT perform on tasks that are genuinely out-of-domain? For example, if MAS-GPT is exclusively trained on mathematics-related tasks (e.g., MATH and GSM8K), would the generated MAS structures remain effective for programming benchmarks such as HumanEval or MBPP? Could you discuss MAS-GPT’s transferability and performance in such scenarios?
3. What are the costs for generating the training data in terms of GPU hours and costs for calling APIs?
4. Have you examined the MAS topologies by representing them as Directed Acyclic Graphs? If so, have you identified any genuinely novel topological structures generated by MAS-GPT beyond those present in the training set? How do you verify that MAS-GPT is not simply memorizing training data topologies and applying them unchanged to tasks during inference?

**Relation To Broader Scientific Literature:**

MAS-GPT builds upon and advances existing literature on multi-agent systems and large language models. It addresses critical limitations found in prior systems such as MetaGPT, ChatDev, and AgentVerse, notably manual configuration and high inference costs. By reframing MAS construction as a generative language task and utilizing SFT, this work bridges a significant gap in current MAS approaches and introduces a flexible, scalable alternative.

**Theoretical Claims:**

Yes

---

> ### Author Rebuttal · Authors · 2025-04-01
>
> We are glad to see that you acknowledge that our approach is novel, methodologically sound, flexible, and scalable. We are sorry that we leave you some concerns and let's clarify.
>
> ---
>
> **W1&Q2:** The generalization capability of MAS-GPT across significantly different or novel task domains remains unclear...
>
> **Answer 1:** Our MAS-GPT can generalize well to new tasks and we have put efforts to verify this point in our paper. Sorry for potentially missing details.
>
> (1) SciQ and SciBench are significantly different, where SciQ is a knowledge-based benchmark (multiple-choice, https://huggingface.co/datasets/allenai/sciq) while SciBench is a reasoning-based benchmark (non-choice, https://huggingface.co/datasets/xw27/scibench). Meanwhile, SciQ is a dataset for **4th grade** exams while SciBench are all **college-level** scientific problems, which are much challenging than those in training data.
>
> (2) Our paper also include results on GPQA (graduate-level QA) and AIME-2024 (mathematical competition), which are all out-of-domain and much harder than training data. Here, we additionally test on medical domain, MedQA (there is no medical dataset in training). From the table, we see that MAS-GPT is indeed generalizable.
> | |GPQA|AIME|MedQA|
> |-|-|-|-|
> |DyLAN|35.98|53.33|76.34|
> |MAS-GPT|37.62|66.67|78.60|
>
> (3) We kindly remind the reviewer that our MAS-GPT has achieved a better generalization capability compared to existing optimization-based methods: MAS-GPT does not require re-optimizing when being applied to different benchmarks. For methods such as GPTSwarm (ICML 2024 oral) and AFlow (ICLR 2025 oral), given a benchmark (e.g., MATH), these methods first optimize on a subset and then could only infer on the corresponding test set.
>
> ---
>
> &nbsp;
>
> **W2:** The generated MAS topologies presented are relatively straightforward...
>
> **A:** Thanks for the comments.
>
> (1) The reason why we develop the MAS topologies at the current complexity is straightforward: the current complexity level is somewhat sufficient to achieve a pleasant performance on most of the benchmarks. All benchmarks are commonly used by MAS community and we follow their setups. If we continue to increase the complexity, it might be hard to achieve a good cost-performnace balance.
>
> (2) Our approach to training MAS-GPT is scalable and methodologically capable of supporting sophisticated topologies. To achieve this, one only needs to design more sophisticated MAS and include them into the MAS pool. Based on this, the trained MAS-GPT would be able to generate sophisticated MAS for complex queries. Meanwhile, MAS-GPT already supports using tools such as code executor, indicating its potential of scalability to include more tools to handle complex tasks.
>
> (3) The main contribution of this paper is pointing out a new direction for the MAS community. Like the training of LLMs (e.g., GPT-4), there are always cases that the trained MAS-GPT cannot solve no matter how well the current MAS-GPT is trained. That is, the current MAS-GPT is not the end. Similar to the continuing advancement of LLMs, our MAS-GPT could be continuously improved as the community designing more sophisticated (or better) MAS, tasks, and data samples.
>
> ---
>
> **Q1:** If two MAS share an identical topology but differ in the instructions to the agent, are they counted as separate MAS instances?
>
> **Answer:** Sorry for the confusion. Yes, they are counted as separate MAS instances. The reasons for such criteria is that it is hard to automatically and merely distinguish topologies.
>
> ---
>
> **Q3:** What are the costs for generating the training data in terms of GPU hours and costs for calling APIs?
>
> **A:** Collecting the training data roughly requires 245k calls of open-source LLMs. The training process takes roughly 32 GPU hours for training 32B-sized models. The refinement process roughly costs 143 US dollars for calling GPT-4o. Please note that MAS-GPT (32B) is only required to be trained **once** and applied to handle diverse queries **without re-training** both MAS-GPT and the LLMs that drive the MAS.
>
> Meanwhile, we will open-source all of the data at every step (e.g., before and after filtering) and models to facilitate future research.
>
> ---
>
> **Q4:** Did you see new topologies?
>
> **Answer:** We have manually checked some of the generated topologies and found that MAS-GPT could generate novel topologies. Please refer to our case study of Case 3 (Section A.3). Meanwhile, please note that even if the generated topologies are the same as existing topologies, MAS-GPT would assign appropriate prompts (instructions) to the agents within the topologies, making them query-specific and appropriate MAS. For example, during the inference of MAS-GPT on GSM-Hard, 925 out of 1000 generated MAS are unseen from the training data, indicating that MAS-GPT is generating query-specific appropriate MAS.
>
> ---
>
> &nbsp;
>
> Overall, we hope that our responses can fully address your concerns and will be grateful for any feedback.

---

### Official Review · Reviewer_y7aF · 2025-03-16

**Overall Recommendation:** 3

**Summary:**

In this paper, the authors propose to train a GPT to generate the code that represents a Multiagent system, and thus provide a team build for each query. Specifically, the author proposes to use some existing datasets as training samples and run these samples on 40 different predefined systems to form training pairs for the model to learn the best candidates. The results on 8 datasets show that the proposed method is better than many fixed baselines, such as CoT and DyLAN.

**Claims And Evidence:**

The motivation of the proposed method is generally clear. The authors want to solve the issue of an adaptive system for each query and human labor, as well as the system cost.

**Essential References Not Discussed:**

The discussion is relatively sufficient.

**Experimental Designs Or Analyses:**

The cost the each sample is not fair. The authors compare the cost by computing the number of the inferences, however, the inference has diverse lengths, which is unfair for short response generators. It is better to include the real money or token cost of the system to have a direct understanding of the result.

The experiment design is unfair. This is two-fold. First, the proposed methods use the training set of the evaluation dataset while the baselines do not use it. Thus, it is the comparison between fine-tuning and zero-shot learning. It is fairer to compare the performance of the cross-domain or zero-shot capability of the proposed method. Second, the baselines are largely included in the system. For instance, the CoT is inside the 40 MAS pool. Thus, as long as the model can figure out the performance of CoT, they can beat it naturally. It will be better not to include the baseline in the pool to see the filter-out performance, mimicking the real-world cases where we do not know what other models are.

**Methods And Evaluation Criteria:**

The motivation for choosing these 40 MAS pools is unclear. Why do you include these models? Does it mean the model can only support existing frameworks? Since the MAS is evolving, the proposed method is more like a model selection algorithm rather than an auto-team building algorithm. Although the system can generalized to unseen MAS systems, it is better to conduct an experiment to enlarge the pool with some random systems already.

**Other Comments Or Suggestions:**

See above.

**Other Strengths And Weaknesses:**

I like the idea of converting the auto-team building tasks to a coding task. And ask GPT to write code to build team.

**Questions For Authors:**

See above.

**Relation To Broader Scientific Literature:**

Highly related to Auto team building or multiagent systems.

**Theoretical Claims:**

N/A

---

> ### Author Rebuttal · Authors · 2025-04-01
>
> Thanks for your appreciation of our idea and motivation. We would like to address your remaining concerns in the following.
>
> ---
>
> &nbsp;
>
> **Methods:** The motivation for choosing these 40 MAS pools is unclear.
>
> **A:** Sorry for the confusion. Let's clarify.
>
> (1) The key motivation for choosing these 40 MAS in the initial pool is for teaching the LLM the basic format for representing a MAS. These 40 MAS cover basic elements such as chain-of-thought prompts, role-playing prompts, LLM-calling functions, and functions to get code execution results. As the base LLM does not know our representation of MAS (verified by poor performance in Figure 5 b, N=0), including these basic elements is critical to teach the LLM our MAS representation.
>
> (2) MAS-GPT is not simply selecting MAS. For example, during the inference of MAS-GPT on GSM-Hard, 925 out of 1000 generated MAS are unseen from the training data, indicating that MAS-GPT is not merely selecting MAS but generating query-specific appropriate MAS.
>
> (3) These 40 MAS constitute only a subset of the full training data. Actually, these 40 MAS only serve as `seed MAS`. They will be evolved during the query-MAS pair refinement process, which indeed enlarges the pool with some random systems.
>
> We will include these in the revision.
>
> ---
>
> &nbsp;
>
> **Exp1:** The authors compare the cost by computing the number of the inferences ... better to include token cost.
>
> **A:** Thanks for the suggestion. Due to limited time, we report the following results and report more in our revision, where we see that MAS-GPT achieves **the best with the least token cost**.
>
> | |AgentVerse|DyLAN|MAS-GPT|
> |-|-|-|-|
> | LLM Calls | 12.05 (70B) | 12.96 (70B) | 1 (32B) + 6.44 (70B)    |
> | Tokens    | 8610 (70B)  | 4874 (70B)  | 1133 (32B) + 2127 (70B) |
> | Acc (%)   | 59.36       | 60.54       | 64.47   |
>
> ---
>
> &nbsp;
>
> **Exp2:** It is fairer to compare the performance of the cross-domain or zero-shot capability.
>
> **A:** Thanks for the suggestion. Acturally, we have compared the performance of the cross-domain or zero-shot capability on datasets such as HumanEval, HumanEval+, GPQA, SciBench, and AIME-2024.
>
> Please note that all of these datasets are **only used for evaluation, but not training**. We would like to highlight that GPQA (graduate-level QA), SciBench (college-level scientific problems), AIME-2024 (mathematical competition) are all much harder than the training data. We also additionally conduct experiments on MedQA (medical domain). We can see that our method achieves **the best performance in these cross-domain setups**. These experiments also verify the generality of our MAS-GPT: it could generalize to domains unseen in training data and to much harder queries than those in training.
>
> | |HumanEval|HumanEval+|GPQA|SciBench|AIME|MedQA|
> |-|-|-|-|-|-|-|
> |DyLAN|79.01|75.78|35.98|19.79|53.33|76.34|
> |MAS-GPT|80.25|78.88|37.62|24.21|66.67|78.60|
>
> ---
>
> &nbsp;
>
> **Exp3:** the baselines are largely included in the system. For instance, the CoT is inside the 40 MAS pool. ... It would be better not to include the baseline in the pool to see the filter-out performance...
>
> **A:** Thanks for the comments.  We would like to answer from three perspectives.
>
> (1) Firstly, our initial MAS pool only includes a few baselines with simple operation such as Co; but does not inlcude complicated baselines such as DyLAN. We include these simple elements to teach the LLMs to generate MAS in our desired format (also see response to methods).
>
> (2) Secondly, we would like to kindly inform the reviewer that including existing baselines into our initial MAS pool does not conflict with the rationale and motivation of MAS-GPT. One exciting and promising advantage of MAS-GPT is that it could `stand on the shoulders of giants`. Ideally, if we could include all existing high-performance MAS methods into the MAS pool, in real-world applications, we only need to deploy MAS-GPT to solve diverse user queries, rather than deploying multiple MAS and designing complicated rules to select them.
>
> (3) Following your suggestions, we exclude CoT from the MAS pool and re-run the experiments. We report the results in the following. From the table, we see that excluding CoT performs comparably with the current version. This result is reasonable because the performance of CoT is normal such that during the process of pair evaluation and selection, CoT is less likely to be paired with queries, resulting in few CoT samples in the training data.
>
> We truly believe in this new paradigm. With more diverse and high-performance MAS being included, we believe that MAS-GPT will be further advanced in a way similar to the advancement of ChatGPT: with better data and training techniques, the models become better.
>
> | |MATH|MMLU|GPQA|
> |-|-|-|-|
> |Before|68.65|78.38|37.62|
> |After|69.09|75.59|37.62|
>
> ---
>
> &nbsp;
>
> Overall, we hope that our responses can fully address your concerns and will be grateful for any feedback.

---

### Decision · Program_Chairs · 2025-05-01

**Decision:**

Accept (poster)

**Comment:**

This paper is concerned with constructing multi-agent systems of LLM agents in an automated way. The idea is to construct a system that can take in a user query and turn it into a multi-agent system that can then be simulated to accomplish some purpose. The authors achieve this by creating a dataset of multi-agent systems suitable for particular user queries. They then adapt their multi-agent system construction LLM by fine tuning on this dataset. The reviewers and I were all in agreement that this is a novel, important, and interesting paradigm. I have myself, in the context of my own work, wondered if such a thing would be possible so it's very nice to see that the authors have gone and done it.

During the review process, a significant issue initially appeared to arise: several reviewers believed that the experimental evaluations in the paper were contaminated by an issue akin to 'training on the test set'. Different reviewers had different versions of this critique. However, it looks to me like the rebuttal responses contained convincing answers to all of them. It is my view that the authors were able to show in the rebuttal that their experimental results did not rely on training on the test set. This conclusion is based on my understanding of a few of the debates that arose between the authors and the reviewers. One debate hinges on the soundness of the authors' "shoulders of giants" argument. I am willing to accept this argument. It seems the real cause of the dispute with some reviewers on this point was more likely a poor explanation for this choice in the paper, so I would recommend the authors work on clarifying this logic in the main text. The other main debate hinged on whether zero-shot generalization was tested and whether the comparison to the baselines was like for like. I believe the authors adequately answered this concern in the rebuttal phase. They pointed specifically to the table in question containing the relevant results.